# Differentially Private Stochastic Optimization: New Results in Convex and Non-Convex Settings

**Raef Bassily**
Department of Computer Science & Engineering
Translational Data Analytics Institute (TDAI)
The Ohio State University
`bassily.1@osu.edu`

**Cristóbal Guzmán**
Department of Applied Mathematics
University of Twente
Inst. for Mathematical and Comput. Eng.
Pontificia Universidad Católica de Chile
`c.guzman@utwente.nl`

**Michael Menart**
Department of Computer Science & Engineering
The Ohio State University
`menart.2@osu.edu`

## Abstract

We study differentially private stochastic optimization in convex and non-convex settings. For the convex case, we focus on the family of non-smooth generalized linear losses (GLLs). Our algorithm for the $\ell_2$ setting achieves optimal excess population risk in near-linear time, while the best known differentially private algorithms for general convex losses run in super-linear time. Our algorithm for the $\ell_1$ setting has nearly-optimal excess population risk $\tilde{O}\big(\sqrt{\frac{\log d}{n\varepsilon}}\big)$, and circumvents the dimension dependent lower bound of [AFKT21] for general non-smooth convex losses. In the differentially private non-convex setting, we provide several new algorithms for approximating stationary points of the population risk. For the $\ell_1$-case with smooth losses and polyhedral constraint, we provide the first nearly dimension independent rate, $\tilde{O}\big(\frac{\log^{2/3} d}{(n\varepsilon)^{1/3}}\big)$ in linear time. For the constrained $\ell_2$-case with smooth losses, we obtain a linear-time algorithm with rate $\tilde{O}\big(\frac{1}{n^{1/3}} + \frac{d^{1/5}}{(n\varepsilon)^{2/5}}\big)$. Finally, for the $\ell_2$-case we provide the first method for *non-smooth weakly convex* stochastic optimization with rate $\tilde{O}\big(\frac{1}{n^{1/4}} + \frac{d^{1/6}}{(n\varepsilon)^{1/3}}\big)$ which matches the best existing non-private algorithm when $d = O(\sqrt{n})$. We also extend all our results above for the non-convex $\ell_2$ setting to the $\ell_p$ setting, where $1 < p \leq 2$, with only polylogarithmic (in the dimension) overhead in the rates.

## 1 Introduction

Stochastic optimization (SO) is a fundamental and pervasive problem in machine learning, statistics and operations research. Here, the goal is to minimize the expectation of a loss function (often referred to as the *population risk*), given only access to a sample of i.i.d. draws from a distribution. When such a sample entails privacy concerns, differential privacy (DP) becomes an important algorithmic desideratum.

Consequently, differentially private stochastic optimization (DP-SO) has been actively investigated for over a decade. Despite major progress in this area, some crucial problems remain with existing

35th Conference on Neural Information Processing Systems (NeurIPS 2021).

| Loss | $\ell_p$-Setting | Rate | Linear Time? | Thm. |
|---|---|---|---|---|
| Convex GLL (Nonsmooth) | $p = 1$ | $\sqrt{\frac{\log d}{n\varepsilon}}$ | | 6 |
| | $p = 2$ | $\max\left(\frac{\sqrt{d}}{n\varepsilon}, \frac{1}{\sqrt{n}}\right)$ | Nearly | 5 |
| Nonconvex Smooth | $p = 1$ | $\frac{\log^{2/3} d}{(n\varepsilon)^{1/3}}$ | $\checkmark$ | 8 |
| | $1 < p \leq 2$ | $\frac{\kappa^{2/3}}{n^{1/3}} + \kappa^{2/3}\left(\frac{d\tilde{\kappa}}{n^2\varepsilon^2}\right)^{1/5}$ | $\checkmark$ | 10 |
| Weakly Convex (Nonsmooth) | $1 \leq p \leq 2$ | $\frac{\kappa^{5/4}}{n^{1/4}} + \kappa^{4/3}\left(\frac{d\tilde{\kappa}}{n^2\varepsilon^2}\right)^{1/6}$ | | 16 |

Table 1: Accuracy bounds and running time for our algorithms. Here, $n$ is sample size, $d$ is dimension, $\varepsilon, \delta$ are the privacy parameters, $\kappa = \min\{\frac{1}{p-1}, \log d\}$ and $\tilde{\kappa} = 1 + \log d \cdot \mathbf{1}(p < 2)$. We omit the dependence on factors of order $\text{polylog}(n, 1/\delta)$. Bounds shown for unit $\ell_p$ ball as a feasible set.

methods. One major problem is the lack of linear-time[1] algorithms for nonsmooth DP-SO (even in the convex case), whereas its non-private counterpart has minimax optimal-risk algorithms which make a single pass over the data [NY83]. A second challenge arises in DP-SCO for non-Euclidean settings; i.e., when the diameter of the feasible set, and Lipschitzness and/or smoothness of losses are measured w.r.t. a non-Euclidean norm (e.g., $\ell_p$ norm). In particular, in the $\ell_1$-setting there is a stark contrast between the polylogarithmic dependence on the dimension in the risk achievable for the smooth case and the necessary polynomial dependence on the dimension in the non-smooth case [AFKT21].

Finally, our understanding of DP-SO in the non-convex case is still quite limited. In the non-convex domain, there are only a few prior results, all of which have several limitations. First, all existing works either assume that the optimization problem is unconstrained or only consider the empirical version of the problem known as differentially private empirical risk minimization (DP-ERM). Obtaining population guarantees based on the empirical risk potentially limits the applicability of the existing methods either in terms of accuracy or in terms of computational efficiency. In particular, all existing methods require super-linear running time w.r.t. the dataset size. Second, most of the existing works consider only the Euclidean setting.[2] Finally, none of the prior works have studied non-convex DP-SO when the loss is non-smooth.

The goal of this work is to provide faster and more accurate methods for DP-SO. Some of the settings we investigate are also novel in the DP literature.

## 1.1 Our Results

We enumerate the different settings we investigate in DP-SO, together with our main contributions.

**Convex generalized linear losses.** Our first case of the study is *non-smooth* DP-SCO in the case of *generalized linear losses* (GLL). This model encompasses a broad class of problems, particularly those which arise in supervised learning, making it a very important particular case. Here, our contributions are two-fold. First, in the $\ell_2$-setting, we provide the first nearly linear-time algorithm that attains the optimal excess risk. The fastest existing methods with similar risk work for general convex losses, but they run in superlinear time w.r.t. sample size [AFKT21, KLL21]. Our second contribution here is a nearly-dimension independent excess risk bound in the $\ell_1$-setting[3] for convex non-smooth GLL. This result circumvents a general DP-SCO excess risk lower bound in the non-smooth $\ell_1$-setting which shows polynomial dependence on the dimension [AFKT21], and it matches the minimax risk in the non-private case when $\varepsilon = \Theta(1)$ [ABRW12].

---

[1]In this work, complexity is measured by the number of gradient evaluations, omitting other operations. This is in line with the oracle complexity model in optimization [NY83].

[2]One exception is [WX19] who study the $\ell_1$ setting in the context of DP-ERM under a fairly strong assumption (see Related Work section).

[3]As in all existing works on DP-SO, in the $\ell_1$-setting we also assume the feasible set to be polyhedral.

Our two contributions for GLL follow the same simple idea. We leverage the GLL structure, namely the fact that these losses are effectively "one-dimensional," to make a fast approximation of the Moreau envelope of the loss [Mor65]. We can then exploit the smoothness of the envelope to improve algorithmic performance. A similar approach was taken by [BFTT19], but their approach suffered from an increase in the running time by a factor of $n^3$ due to the high cost of approximating the gradient of the envelope, which involves solving a high dimensional strongly convex optimization problem at each iteration. In the case of $\ell_2$, we use an existing linear-time algorithm for smooth DP-SCO with optimal excess risk [FKT20] combined with our smoothing approach, which results in an $O(n \log n)$-time algorithm. In the case of $\ell_1$, we use an existing noisy Frank-Wolfe algorithm that attains *optimal empirical risk* for *smooth losses* [TTZ15], together with generalization bounds for GLLs based on Rademacher complexity [SSBD14]. This algorithm is not linear time, and hence it is tempting to instead use a variant of one pass stochastic Frank-Wolfe algorithms, as in [AFKT21, BGN21]. However, the excess risk of these algorithms has a linear dependence on the smoothness constant, which prevents us from obtaining the optimal risk via smoothing. Hence, it is an interesting future direction to improve the running time in the $\ell_1$-setting.

**Non-convex Smooth Losses.** Next, we move to the setting of smooth non-convex losses, where the goal is to approximate *first-order stationary points*[4] (see (1) in Section 2). This case has attracted significant attention recently, and it brings major theoretical challenges since most tools used to derive optimal excess risk in DP-SCO, such as uniform stability [HRS16, BFGT20] or privacy amplification by iteration [FKT20], no longer apply. Here, we provide the first linear time private algorithms. In the $\ell_1$-setting, we obtain a nearly-dimension independent rate $O((\log^2 d/[n\varepsilon])^{1/3})$, which to the best of our knowledge is new, even in the non-private case. We suspect that our rates for the $\ell_1$-setting are essentially tight for linear-time algorithms (at least when $\varepsilon = \Theta(1)$). In [ACD+19], for non-convex smooth SO in the $\ell_2$-setting an oracle complexity lower bound $\Omega(\alpha^{-3})$ is shown for attaining $\alpha$-approximate stationary points via a stochastic gradient oracle. This, together with the non-private upper bound [FLLZ18, HKMS20, ZSM+20], implies that in the non-private setting, the stationarity rate $1/n^{1/3}$ is optimal for linear-time stochastic first-order algorithms. In the $\ell_2$-setting (and more generally, for $\ell_p$-setting, where $1 \le p \le 2$) our stationarity rate (see Table 1) is slightly worse than the state of the art, $O((d/n^2)^{1/4})$ [ZCH+20]. However, in [ZCH+20], only the unconstrained case is considered, where the accuracy measure is the norm of the gradient; moreover, the running time is superlinear, $O(n^2\varepsilon/\sqrt{d})$.

Our workhorse for these results is a recently developed variance-reduced stochastic Frank-Wolfe method [HKMS20, ZSM+20], which has also proved useful in DP-SCO [AFKT21, BGN21]. This method is based on reducing variance through a recursive estimate of the gradient at the current point, leveraging past gradient estimates and the fact that step-sizes are small. Applying this technique in DP is challenging, as we need to carefully schedule the algorithm in rounds (to prevent gradient error accumulation) and to properly tune step-sizes and noise, in order to trade-off accuracy and privacy.

**Non-convex non-smooth losses.** We conclude with the case of weakly convex non-smooth stochastic optimization, where we devise algorithms to compute *close to nearly-stationary points*. Weakly convex functions are a natural and rather common model in some machine learning applications, including convex composite losses, robust phase retrieval, non-smooth trimmed estimation, covariance matrix estimation, sparse dictionary learning, etc. (see [DG19, DD19] and references therein). Moreover, this class subsumes smooth non-convex functions. To the best of our knowledge, this setting has not been previously addressed in the DP literature. Our algorithm is inspired by the proximally-guided stochastic subgradient method from [DG19], and it is based on approximating proximal steps w.r.t. the risk function, where each proximal subproblem is solved through an optimal DP-SCO method for strongly convex losses [AFKT21]. This algorithm works similarly for the $\ell_1$ and $\ell_2$ settings (and, in fact, $\ell_p$ for any $1 \le p \le 2$), for which we exploit the strong convexity properties of these spaces. Here again, our non-Euclidean extensions seem to be new, even in the non-private case. Our rates for $\ell_2$-setting match the best existing non-private rates, $O(1/n^{1/4})$, in the regime $d = O(\sqrt{n})$ (when $\varepsilon = \Theta(1)$). Finally, we observe that our algorithm runs in time $\tilde{O}(\min\{n^{3/2}, n^2\varepsilon/\sqrt{d}\})$.

---

[4]Unless otherwise stated, we will refer to first-order stationary points as stationary points.

## 1.2 Related Work

Differentially private convex optimization has been studied extensively for over a decade (see, e.g., [CMS11, JKT12, KST12, BST14, JT14, TTZ15, BFTT19, FKT20]). Most of the early works in this area focused on the empirical risk minimization problem. The first work to derive minimax optimal excess risk in DP-SCO is [BFTT19], which has been further improved, in terms of running time (e.g. [FKT20, BFGT20, KLL21]). Non-Euclidean settings in DP convex optimization were studied in [JKT12, TTZ15]. Nearly optimal rates for non-Euclidean DP-SCO were only recently discovered in [AFKT21, BGN21]. [JT14] was one of the first works to focus on the case of private optimization for GLLs, and showed that dimension independent excess risk was possible in $\ell_1$ and $\ell_2$ settings. These results have since been superseded in the $\ell_1$ case by [AFKT21] and in the $\ell_2$ case by [SSTT21].

In the non-convex case, [ZZMW17, WYX17, WJEG19] studied smooth unconstrained DP-ERM in the Euclidean setting. Smooth unconstrained DP-SO was studied in [WCX19], where relatively weak guarantees on the excess risk were shown. Convergence to second-order stationary points of the empirical risk was also studied in the same reference under stronger smoothness assumptions. Smooth constrained DP-ERM was studied in [WX19] in both $\ell_2$ and $\ell_1$ settings. However, their result in the $\ell_1$ setting entails the strong assumption that the loss is smooth w.r.t. the $\ell_2$ norm. The special case of non-convex smooth GLLs was studied in [SSTT21], however, their result is limited to the empirical risk (DP-ERM) in the unconstrained setting. The work of [ZCH+20] studied DP-SO in the Euclidean setting, and gave convergence guarantees in terms of the population gradient, however, their results are limited to smooth unconstrained optimization.

## 2 Preliminaries

**Normed Spaces.** Let $(\mathbf{E}, \|\cdot\|)$ be a normed space of dimension $d$, and let $\langle\cdot,\cdot\rangle$ an arbitrary inner product over $\mathbf{E}$ (not necessarily inducing the norm $\|\cdot\|$). Given $x \in \mathbf{E}$ and $r > 0$, let $\mathcal{B}_{\|\cdot\|}(x,r) = \{y \in \mathbf{E} : \|y - x\| \leq r\}$. The dual norm over $\mathbf{E}$ is defined as usual, $\|y\|_* \triangleq \max_{\|x\| \leq 1}\langle y, x\rangle$. With this definition, $(\mathbf{E}, \|\cdot\|_*)$ is also a $d$-dimensional normed space. As a main example, consider the case of $\ell_p^d \triangleq (\mathbb{R}^d, \|\cdot\|_p)$, where $1 \leq p \leq \infty$ and $\|x\|_p \triangleq \left(\sum_{j\in[d]}|x_j|^p\right)^{1/p}$. As a consequence of the Hölder inequality, one can prove that the dual of $\ell_p^d$ corresponds to $\ell_q^d$, where $1 \leq q \leq \infty$ is the conjugate exponent of $p$, determined by $1/p + 1/q = 1$.

**Differential Privacy [DKM+06].** A randomized algorithm $\mathcal{A}$ is said to be $(\varepsilon, \delta)$ differentially private (abbreviated $(\varepsilon, \delta)$-DP) if for any pair of datasets $S$ and $S'$ differing in one point and any event $\mathcal{E}$ in the range of $\mathcal{A}$ it holds that

$$\mathbb{P}[\mathcal{A}(S) \in \mathcal{E}] \leq e^\varepsilon \mathbb{P}[\mathcal{A}(S') \in \mathcal{E}] + \delta.$$

**Lemma 1** (Advanced composition [DRV10, DR14]). *For any $\varepsilon > 0, \delta \in [0,1)$, and $\delta' \in (0,1)$, the class of $(\varepsilon, \delta)$-differentially private algorithms satisfies $(\varepsilon', k\delta + \delta')$-differential privacy under k-fold adaptive composition, for $\varepsilon' = \varepsilon\sqrt{2k\log(1/\delta')} + k\varepsilon(e^\varepsilon - 1)$.*

**Stochastic Optimization.** In the Stochastic Optimization problem with $(\mathbf{E}, \|\cdot\|)$-setting, we have a normed space $(\mathbf{E}, \|\cdot\|)$; a feasible set $\mathcal{W} \subseteq \mathbf{E}$ which is closed, convex and with diameter at most $D$ w.r.t. $\|\cdot\|$; and loss functions $f : \mathcal{W} \times \mathcal{Z} \mapsto \mathbb{R}$ are assumed to be $L_0$-Lipschitz w.r.t. $\|\cdot\|$. Sometimes, we also consider losses which are $L_1$-smooth: i.e., for all $w, v \in \mathcal{W}$, $\|\nabla f(w) - \nabla f(v)\|_* \leq L_1\|w - v\|$. In this problem, there is an unknown distribution $\mathcal{D}$ over a set $\mathcal{Z}$, and our goal is to minimize a certain accuracy measure that depends on the population risk, defined as $F_{\mathcal{D}}(w) = \mathbb{E}_{z\sim\mathcal{D}}[f(w, z)]$, when only given access to a sample $S = (z_1, ..., z_n) \overset{i.i.d.}{\sim} \mathcal{D}$. In Differentially Private Stochastic Optimization (DP-SO) one is concerned with solving this problem under the constraint that the algorithm used is $(\varepsilon, \delta)$-DP w.r.t. $S$.

Depending on additional assumptions of the losses, the accuracy measure in DP-SO may vary. In the *convex case*, the accuracy of a stochastic optimization algorithm is naturally measured by the excess population risk, defined as $F_{\mathcal{D}}(w) - \min_{v\in\mathcal{W}} F_{\mathcal{D}}(v)$. For the non-convex case, providing guarantees on the excess population risk is often intractable.

**Non-Convex Stochastic Optimization.** In the *non-convex smooth* case, a common performance measure to use is the *stationarity gap* of the population risk, which for $w \in \mathcal{W}$ is defined as

$$\mathsf{Gap}_{F_\mathcal{D}}(w) = \max_{v \in \mathcal{W}} \langle \nabla F_\mathcal{D}(w), w - v \rangle. \tag{1}$$

Note that if the stationarity gap is zero, then $w$ is indeed a stationary point of the risk. For the *non-convex non-smooth* case, near stationarity (i.e., small stationarity gap) is often a stringent concept, as the set of points with small stationarity gap may coincide with the stationary points themselves. Hence, we will consider instead the goal of finding *close to nearly-stationary points* [DG19, DD19], which we formally introduce in Section 5.

## 3 Algorithms for Convex Non-smooth Generalized Linear Losses

In this section we consider the case when $f$ is a non-smooth generalized linear loss.

**Definition 2** (Generalized Linear Loss). We say that $f : \mathcal{W} \times (\mathcal{X} \times \mathbb{R}) \to \mathbb{R}$ is an $L_0$-Lipschitz, $R$-bounded GLL with respect to norm $\|\cdot\|$ if $\max_{x \in \mathcal{X}} \|x\|_* \leq R$ and for every $y \in \mathbb{R}$ there exists a function $\ell^{(y)} : \mathbb{R} \to \mathbb{R}$ such that $f(w, (x, y)) = \ell^{(y)}(\langle x, w \rangle)$ and $\ell^{(y)}$ is $L_0$-Lipschitz.

We will occasionally refer to the $x$ component of a datapoint as the feature vector. Note the GLL definition implies that $f(\cdot, z)$ is $(L_0 R)$-Lipschitz. By smoothing the function $f$ through $\ell$, one can obtain a smoothing which is both efficient and invariant to the norm. The first property can be used to attain an optimal rate for DP-SCO in nearly linear time. The later property allows for an essentially optimal, nearly dimension independent rate in the $\ell_1$ setting for *non-smooth* GLLs.

### 3.1 Smoothing Generalized Linear Losses

Existing works such as [BFTT19] have used the Moreau envelope smoothing [Mor65] for DP-SCO, but suffer from the high computational cost of computing the proximal operator. For GLLs, we can smooth $\ell$ instead of $f$ to obtain a smoothed function efficiently. We leave the details of the Moreau smoothing and proofs for Appendix A.1 and focus here on our results. We have the following guarantee for the smoothed version of $f$.

**Lemma 3.** Let $(x, y) \in (\mathcal{X} \times \mathbb{R})$. Let $\ell_\beta^{(y)}$ be the Moreau envelope of $\ell^{(y)}$ and define $f_\beta(w, z) = \ell_\beta^{(y)}(\langle w, x \rangle)$. Then $f_\beta$ is $2L_0 R$-Lipschitz and $\beta\|x\|_*^2$-smooth with respect to $\|\cdot\|$ and $|f(w, (x, y)) - f_\beta(w, (x, y))| \leq \frac{2L_0^2}{\beta}$ for all $w \in \mathcal{W}$.

By smoothing $f$ through $\ell$, we reduce the evaluation of the proximal operator to a 1-*dimensional convex problem*. This allows us to use the bisection method to obtain the following oracle for $f_\beta$ which runs in logarithmic time.

**Lemma 4.** Let $\beta, \alpha > 0$ and let $\|\cdot\|$ be a norm. Then the there exists a gradient oracle, $\mathcal{O}_{\beta, \alpha, R}$ for $f_\beta$ (Algorithm 3 in Appendix A.1) which satisfies $\|\nabla f_\beta(w, (x, y)) - \mathcal{O}_{\beta, \alpha, R}(w, (x, y))\|_* \leq \alpha$ for any $x$ such that $\|x\|_* \leq R$. Further, $\mathcal{O}_{\beta, \alpha, R}$ has running time $O\left(\log(L_0^2 R^2 / \alpha^2)\right)$.

### 3.2 New Results from Smoothing

**Linear Time DP-SCO in the $\ell_2$ Setting.** Given the oracle described above, we can optimize $f_\beta$ using the linear time Phased-SGD algorithm of [FKT20]. When using $\mathcal{O}_{\beta, \alpha, R}$ instead of the true gradient oracle, $\nabla f$, we need account for two additive penalties, the increase in error due to using the approximate gradient and the increase in error to due to minimizing the smoothed function (see Appendix A.2 for details). We ultimately have the following guarantee.

**Theorem 5.** Let $\mathcal{W} \subset \mathbb{R}^d$ have $\|\cdot\|_2$-diameter at most $D$. Let $f : \mathcal{W} \times (\mathcal{X} \times \mathbb{R}) \to \mathbb{R}$ be a $L_0$-Lipschitz and $R$-bounded GLL with respect to $\|\cdot\|_2$. Let $\beta = \sqrt{n}L_0/R$, $\alpha = \frac{L_0 R}{n \log n}$. Then Phased-SGD run with oracle $\mathcal{O}_{\beta, \alpha, R}$ and dataset $S \in \mathcal{Z}^n$ satisfies $(\varepsilon, \delta)$ differential privacy and has running time $O(n \log n)$. Further, if $S \sim \mathcal{D}^n$ the output of Phased-SGD has expected excess population risk $O\left(L_0 RD \left(\frac{\sqrt{d \log(1/\delta)}}{n\varepsilon} + \frac{1}{\sqrt{n}}\right)\right)$.

**Remark.** *The algorithm which implements $\mathcal{O}_{\beta,\alpha,R}$ (Algorithm 3 in Appendix A.1) also works in the unconstrained case ($\mathcal{W} = \mathbb{R}^d$). This is due to the fact that evaluating the gradient of the smoothed function involves solving a regularized objective, which naturally bounds the region in which the minimizer lies. For differentially private GLLs this is relevant, as [SSTT21] showed that in this setting one can achieve dimension independent excess risk. For more details, see Appendix A.2.*

**Better Rate for $\ell_1$ Setting.** Another interesting consequence of the above smoothing is that, because it is scalar in nature, it allows us to achieve better rates in the $\ell_1$-setting. In [AFKT21] it was shown that the optimal rate for general non-smooth losses under $(\varepsilon, \delta)$-DP was roughly $\Omega(\sqrt{d}/[n\varepsilon \log d])$. However, their lower bound does not apply to GLLs. In the following, we show that using the smoothing technique previously described we can achieve a better rate of $\tilde{O}(1/\sqrt{n\varepsilon})$. We note this rate is optimal in the regime $\varepsilon = \Theta(1)$ [ABRW12].

**Theorem 6.** *Let $\mathcal{W} \subset \mathbb{R}^d$ be a polytope defined by a set of vertices $\mathcal{V}$ of cardinality $J$, where $\mathcal{W} = Conv(\mathcal{V})$ and $\mathcal{W}$ has $\|\cdot\|_1$-diameter at most $D$. Let $f : \mathcal{W} \times (\mathcal{X} \times \mathbb{R}) \to \mathbb{R}$ be a $L_0$-Lipschitz and R-bounded GLL with respect to $\|\cdot\|_1$. Let $\beta = \frac{L_0\sqrt{n\varepsilon}}{RD\log^{1/4}(1/\delta)\sqrt{\log(J)\log(n)}}$ and $\alpha = \frac{1}{n\log(n)}$. Then Noisy Frank Wolfe (Algorithm 5 in Appendix A.3) with oracle $\mathcal{O}_{\beta,\alpha,R}$ and dataset $S \in \mathcal{Z}^n$ satisfies $(\varepsilon, \delta)$-differential privacy. Further, if $S \sim \mathcal{D}^n$ the output of Noisy Frank Wolfe has expected excess population risk $O\left(L_0 RD\left(\frac{\log^{1/4}(1/\delta)\sqrt{\log(J)\log(n)}}{\sqrt{n\varepsilon}} + \frac{\sqrt{\log d}}{\sqrt{n}}\right)\right)$.*

Proof details are located in Appendix A.3.

## 4 Algorithms for Non-convex Smooth Losses

In this section, we describe differentially private algorithms for non-convex smooth stochastic optimization in the $\ell_p$-setting for $1 \leq p \leq 2$. We provide formal convergence guarantees in terms of the stationarity gap (see (1) in Section 2). Our algorithms are inspired by the variance-reduced stochastic Frank-Wolfe algorithm [ZSM+20]. However, our algorithms involve several crucial differences from their non-private counterpart. In particular, they are divided into a number of rounds $R = O(\log(n))$, where each round $r \in \{0, \ldots, R-1\}$ involves $2^r$ updates for the iterate. Each round $r$ starts by computing a fresh estimate for the gradient of the population risk at the current iterate based on a large batch of data points, then such gradient estimate is updated recursively using disjoint batches of decreasing size sampled across the $2^r$ iterations of that round. Using this round-based structure and batch schedule, together with carefully tuned step sizes, allows us to effectively control the privacy budget while attaining small stationarity gap w.r.t. the population risk. Moreover, our algorithms make a *single pass* on the input sample, i.e., they run in linear time.

In this section, we assume that $\forall z \in \mathcal{Z}$, $f(\cdot, z)$ is $L_0$-Lipschitz and $L_1$-smooth loss in the respective $\ell_p$ norm. A key tool we use in our analysis in this section is the notion of *regularity of normed spaces* [JN08]. Roughly speaking, this notion captures the smoothness properties of the dual norm, which allows us to bound the error in the gradient estimates computed across the iterations of our algorithms. This, in turn, allows us to bound the stationarity gap of the output. In fact, this notion of regularity extends the applicability of our algorithms to general spaces whose dual has a sufficiently smooth norm. For more details on this notion, we refer the reader to Appendix B.

### 4.1 Algorithm for Polyhedral and $\ell_1$ Settings

We consider the *polyhedral* setup, namely, we consider a normed space $(\mathbf{E}, \|\cdot\|)$, where the unit ball w.r.t. the norm, $\mathcal{B}_{\|\cdot\|}$ is a convex polytope with at most $J$ vertices. The feasible set $\mathcal{W}$, is a polytope with at most $J$ vertices and $\|\cdot\|$-diameter $D > 0$.

The formal guarantees of Algorithm 1 are stated below. Detailed proofs can be found in Appendix B.1.

**Theorem 7.** *Let $\eta_{r,t} = \frac{1}{\sqrt{t+1}}$ $\forall r, t$. Then, Algorithm 1 is $(\varepsilon, \delta)$-differentially private.*

---

**Algorithm 1** $\mathcal{A}_{\mathsf{polySFW}}$: Private Polyhedral Stochastic Frank-Wolfe Algorithm

---

**Require:** Dataset $S = (z_1, \ldots, z_n) \in \mathcal{Z}^n$, privacy parameters $(\varepsilon, \delta)$, polyhedral set $\mathcal{W}$ with $J$ vertices $\mathcal{V} = (v_1, \ldots, v_J)$, number of rounds $R$, batch size $b$, step sizes $(\eta_{r,t} : r = 0, \ldots, R-1, \ t = 0, \ldots, 2^r - 1)$.

1: Choose an arbitrary initial point $w_0^0 \in \mathcal{W}$
2: **for** $r = 0$ to $R - 1$ **do**
3:     Let $s_r = 2D(L_0 + L_1 D)\frac{2^r \sqrt{\log(1/\delta)}}{b\varepsilon}$
4:     Draw a batch $B_r^0$ of $b$ samples without replacement from $S$
5:     Compute $\nabla_r^0 = \frac{1}{b}\sum_{z \in B_r^0} \nabla f(w_r^0, z)$
6:     $v_r^0 = \arg\min_{v \in \mathcal{V}} \{\langle v, \nabla_r^0 \rangle + u_r^0(v)\}$, where $u_r^0(v) \sim \mathsf{Lap}(s_r)$
7:     $w_r^1 \leftarrow (1 - \eta_{r,0})w_r^0 + \eta_{r,0}v_r^0$
8:     **for** $t = 1$ to $2^r - 1$ **do**
9:         Draw a batch $B_r^t$ of $b/(t+1)$ samples without replacement from $S$.
10:       Compute $\Delta_r^t = \frac{t+1}{b}\sum_{z \in B_r^t}\left(\nabla f(w_r^t, z) - \nabla f(w_r^{t-1}, z)\right)$
11:       $\nabla_r^t = (1 - \eta_{r,t})\left(\nabla_r^{t-1} + \Delta_r^t\right) + \eta_{r,t}\frac{t+1}{b}\sum_{z \in B_r^t}\nabla f(w_r^t, z)$
12:       Compute $v_r^t = \arg\min_{v \in \mathcal{V}}\langle v, \nabla_r^t \rangle + u_r^t(v)$, where $u_r^t(v) \sim \mathsf{Lap}(s_r)$
13:       $w_r^{t+1} \leftarrow (1 - \eta_{r,t})w_r^t + \eta_{r,t}v_r^t$
14:     $w_{r+1}^0 = w_r^{2^r}$
15: Output $\widehat{w}$ uniformly chosen from $\left(w_r^t : r \in \{0, \ldots, R-1\}, t \in \{0, \ldots, 2^r - 1\}\right)$

---

**Theorem 8.** *Let* $R = \frac{2}{3}\log\left(\frac{n\varepsilon}{\log^2(J)\log^2(n)\sqrt{\log(1/\delta)}}\right)$, $b = \frac{n}{\log^2(n)}$, *and* $\eta_{r,t} = \frac{1}{\sqrt{t+1}} \ \forall r, t$. *Let* $\mathcal{D}$ *be any distribution over* $\mathcal{Z}$. *Let* $S \sim \mathcal{D}^n$ *be the input dataset. The output* $\widehat{w}$ *of Algorithm 1 satisfies*

$$\mathbb{E}\left[\mathsf{Gap}_{F_{\mathcal{D}}}(\widehat{w})\right] = O\left(D(L_0 + L_1 D) \cdot \frac{\log^{2/3}(J)\log^{2/3}(n)\log^{1/6}(1/\delta)}{n^{1/3}\varepsilon^{1/3}}\right).$$

The proof of the above theorem relies on the following lemma, which gives a bound on the expected error in the gradient estimates $\nabla_r^t$.

**Lemma 9.** *Let* $\mathcal{D}$ *be any distribution over* $\mathcal{Z}$. *Let* $S \sim \mathcal{D}^n$ *be the input dataset of Algorithm 1. Let the step sizes* $\eta_{r,t} = \frac{1}{\sqrt{t+1}} \ \forall r, t$. *For every* $r \in \{0, \ldots, R-1\}$, $t \in \{0, \ldots, 2^r - 1\}$, *the recursive gradient estimator* $\nabla_r^t$ *satisfies*

$$\mathbb{E}\left[\|\nabla_r^t - \nabla F_{\mathcal{D}}(w_r^t)\|_*\right] \leq 4L_0\sqrt{\frac{\log(J)}{b}}\left(1 - \frac{1}{\sqrt{t+1}}\right)^{t+1} + 4(L_1 D + L_0)\frac{\log(J)}{\sqrt{b}}(t+1)^{1/4}.$$

### 4.2 Algorithm for $\ell_p$ Settings when $1 < p \leq 2$

Our algorithm in this setting has a similar structure to Algorithm 1 in Section 4.1, except for the following few, but crucial, differences. First, for all iterations $(r, t)$: the recursive gradient estimate $\nabla_r^t$ and the gradient variation estimate $\Delta_r^t$ are replaced with noisy versions $\widetilde{\nabla}_r^t$ and $\widetilde{\Delta}_r^t$ obtained by adding Gaussian noise to the respective quantities. Namely, in each round $r = 0, \ldots, R-1$, Steps 5, 10, and 11 in Algorithm 1 are now, respectively, computed as follows:

$\widetilde{\nabla}_r^0 = \frac{1}{b}\sum_{z \in B_r^0}\nabla f(w_r^0, z) + N_r^0$, $\quad \widetilde{\Delta}_r^t = \frac{t+1}{b}\sum_{z \in B_r^t}\left(\nabla f(w_r^t, z) - \nabla f(w_r^{t-1}, z)\right) + \widehat{N}_r^t$, and

$\widetilde{\nabla}_r^t = (1 - \eta_{r,t})\left(\widetilde{\nabla}_r^{t-1} + \widetilde{\Delta}_r^t\right) + \eta_{r,t}\left(\frac{t+1}{b}\sum_{z \in B_r^t}\nabla f(w_r^t, z) + N_r^t\right)$ where $N_r^t \sim \mathcal{N}\left(0, \sigma_{r,t}^2 \mathbb{I}_d\right)$

and $\widehat{N}_r^t \sim \mathcal{N}\left(0, \widehat{\sigma}_{r,t}^2 \mathbb{I}_d\right)$. The noise parameters are chosen such that the algorithm is $(\varepsilon, \delta)$-DP.

In particular, we set $\sigma_{r,t}^2 = \frac{16L_0^2(t+1)^2 d^{\frac{2}{p}-1}\log(1/\delta)}{b^2\varepsilon^2}$ and $\widehat{\sigma}_{r,t}^2 = \frac{16L_1^2 D^2 \eta_{r,t}^2(t+1)^2 d^{\frac{2}{p}-1}\log(1/\delta)}{b^2\varepsilon^2}$. The second difference here pertains to the way the iterates are updated, which now becomes $w_r^{t+1} = (1 - \eta_{r,t})w_r^t + \eta_{r,t}\arg\min_{v \in \mathcal{W}}\langle v, \widetilde{\nabla}_r^t \rangle$. Finally, we use a different setting for the number of rounds $R$ than the one used earlier. Below, we state the formal guarantees of this algorithm, which we refer to as *noisy stochastic Frank-Wolfe* $\mathcal{A}_{\mathsf{nSFW}}$. In Appendix B.2, we give a formal description of this algorithm (Algorithm 6) together with full proofs of the statements below.

**Theorem 10.** *Algorithm $\mathcal{A}_{\mathsf{nSFW}}$ (Algorithm 6 in Appendix B.2) is $(\varepsilon, \delta)$-DP.*

**Theorem 11.** *Consider the $\ell_p$ setting of non-convex smooth stochastic optimization, where $1 < p \leq 2$. Let $\kappa = \min\left(\frac{1}{p-1}, 2\log(d)\right)$ and $\widetilde{\kappa} = 1 + \log(d) \cdot \mathbf{1}(p < 2)$. In $\mathcal{A}_{\mathsf{nSFW}}$, let $R = \frac{4}{5}\log\left(\frac{n\varepsilon}{\sqrt{d\widetilde{\kappa}\log(1/\delta)}\,\kappa^{5/3}\log^2(n)}\right)$, $b = \frac{n}{\log^2(n)}$, and $\eta_{r,t} = \frac{1}{\sqrt{t+1}}$ $\forall r, t$. Let $\mathcal{D}$ be any distribution over $\mathcal{Z}$, and $S \sim \mathcal{D}^n$ be the input dataset. The output $\widehat{w}$ satisfies:*

$$\mathbb{E}\left[\mathsf{Gap}_{F_{\mathcal{D}}}(\widehat{w})\right] = O\left(D(L_0 + L_1 D)\kappa^{2/3}\left(\frac{\log^{2/3}(n)}{n^{1/3}} + \frac{d^{1/5}\widetilde{\kappa}^{1/5}\log^{1/5}(1/\delta)\log^{4/5}(n)}{n^{2/5}\varepsilon^{2/5}}\right)\right).$$

*Note that for the Euclidean setting, we have $\kappa = \widetilde{\kappa} = 1$ in the above bound.*

The proof of the above theorem has a similar outline to that of Theorem 8 with a few exceptions to account for the additional noise in the gradient estimates $\widetilde{\nabla}_r^t$. As before, the proof of this theorem relies on the following lemma that bounds the error in the gradient estimates in the dual norm.

**Lemma 12.** *Let $\mathcal{D}$ be any distribution over $\mathcal{Z}$, and $S \sim \mathcal{D}^n$ be the input dataset. For the same settings of parameters in Theorem 11, the gradient estimate $\widetilde{\nabla}_r^t$ satisfies the following for all $r, t$:*

$$\mathbb{E}\left[\left\|\widetilde{\nabla}_r^t - \nabla F_{\mathcal{D}}(w_r^t)\right\|_*\right] \leq 8L_0\left(\sqrt{\frac{\kappa}{b}} + \frac{\sqrt{d\kappa\widetilde{\kappa}\log(1/\delta)}}{b\varepsilon}\right)\left(1 - \frac{1}{\sqrt{t+1}}\right)^{t+1}$$

$$+ 16\left(L_1 D + L_0\right)\left(\frac{\kappa}{\sqrt{b}}(t+1)^{1/4} + \frac{\sqrt{d\kappa\widetilde{\kappa}\log(1/\delta)}}{b\varepsilon}(t+1)^{3/4}\right).$$

## 5 Algorithm for Weakly Convex Non-smooth Losses

Our final setting is DP stochastic *weakly convex* optimization. Much of the theory of weakly-smooth functions is available in [RW98], but we provide a self-contained exposition in Appendix C.1.[5] We recall that a function $f : \mathcal{W} \mapsto \mathbb{R}$ is $\rho$-weakly convex w.r.t. $\|\cdot\|$ if for all $0 \leq \lambda \leq 1$ and $w, v \in \mathcal{W}$,

$$f(\lambda w + (1-\lambda)v) \leq \lambda f(w) + (1-\lambda)f(v) + \frac{\rho\lambda(1-\lambda)}{2}\|w - v\|^2. \tag{2}$$

It is easy to see that any $L_1$-smooth function is indeed $L_1$-weakly convex, so weak convexity encompasses smooth non-convex functions (see Corollary 25 in Appendix C.1). However, this extension is interesting as it also contains some classes of non-smooth functions.

### 5.1 Proximal-Type Operator and Proximal Near Stationarity

The next property is crucial for regularization of weakly smooth functions, and it would allow us to make sense of a proximal-type operator in some non-Euclidean norms. Proofs of results in this section can be found in Appendix C.2.

**Proposition 13.** *Let $\|\cdot\|$ be a norm such that $\frac{1}{2}\|\cdot\|^2$ is $\nu$-strongly convex w.r.t. $\|\cdot\|$. If $f$ is $\rho$-weakly convex and $\nu\beta \geq \rho$, then the function $w \mapsto f(w) + \frac{\beta}{2}\|w - u\|^2$ is $(\nu\beta - \rho)$-strongly convex w.r.t. $\|\cdot\|$.*

We provide now some useful results regarding a proximal-type mapping for weakly convex functions in normed spaced. This provides a non-Euclidean counterpart to results in [RW98, DG19, DD19]. First, given $\mathcal{W} \subseteq \mathbf{E}$ a closed and convex set, we define the proximal-type mapping as:

$$\mathsf{prox}_f^\beta(w) = \arg\min_{v \in \mathcal{W}}\left[f(v) + \frac{\beta}{2}\|v - w\|^2\right]. \tag{3}$$

Despite the stark similarity with the Euclidean proximal operator, the characterization of proximal points is in general different (due to the formula for the subdifferential of the squared norm), so we need to re-derive some near-stationarity estimates derived in [DD19, DG19].

---

[5]Our motivation to reproduce the basic theory stems from the fact that [RW98] and much of the literature of weakly convex functions focuses on Euclidean settings, whereas we are interested in more general $\ell_p$ settings.

**Lemma 14.** *Let $\| \cdot \|$ be such that $\frac{1}{2}\| \cdot \|^2$ is differentiable and $\nu$-strongly convex w.r.t. $\| \cdot \|$, let $f : \mathbf{E} \mapsto \mathbb{R}$ be a $\rho$-weakly convex subdifferentiable function, $\mathcal{W} \subseteq \mathbf{E}$ a closed, convex set with diameter $D$, and $\beta > \rho/\nu$. Then, for any $w \in \mathcal{W}$, the proximal-type mapping $\hat{w} = prox_f^\beta(w)$ (given in (3)) is well-defined, and moreover there exists $g \in \partial f(\hat{w})$ such that*

$$\sup_{v \in \mathcal{W}} \langle g, \hat{w} - v \rangle \leq \beta D \| w - \hat{w} \|.$$

The previous lemma is the key insight on the accuracy guarantee and algorithms we will use for stochastic weakly convex optimization. First, note that in the weakly convex setting it is unlikely to find points with small norm of the gradient or small stationarity gap; however, we will settle for points $w \in \mathcal{W}$ which are $\vartheta$-*close to a nearly-stationary point* [DD19, DG19], i.e., that satisfies

$$(\exists \hat{w} \in \mathcal{W})(\exists g \in \partial f(\hat{w})) : \quad \| w - \hat{w} \| \leq \vartheta \quad \text{and} \quad \sup_{v \in \mathcal{W}} \langle g, \hat{w} - v \rangle \leq \vartheta. \tag{4}$$

Above, $\vartheta \geq 0$ is the accuracy parameter. This accuracy measure states that $w$ is at distance at most $\vartheta$ from a $\vartheta$-nearly stationary point. It is then apparent how the proximal-type operator can certify (4). For convenience, we define a notion of efficiency in weakly-convex DP-SO, particularly geared towards algorithms that certify close to near stationarity via the proximal-type mapping.

**Definition 15** (Proximal Near Stationarity). A randomized algorithm $\mathcal{A} : \mathcal{Z}^n \mapsto \mathbf{E}$, for the stochastic optimization problem $\min_{w \in \mathcal{W}} F_{\mathcal{D}}(w)$, achieves $(\vartheta, \beta)$-proximal near stationarity if

$$\mathbb{E}_{S \sim \mathcal{D}^n, \mathcal{A}} \big[ \| prox_{F_{\mathcal{D}}}^\beta(\mathcal{A}(S)) - \mathcal{A}(S) \| \big] \leq \vartheta / \max\{1, \beta D\}. \tag{5}$$

Notice the maximum in the denominator is a normalizing factor, inspired by Lemma 14. Note further that, by Lemma 14, an algorithm with proximal near stationarity ensures closeness to nearly stationary points through its proximal-type mapping: namely, if $\mathcal{A}$ satisfies Definition 15, then

$$\mathbb{E}_{S \sim \mathcal{D}^n, \mathcal{A}} \big[ \| prox_{F_{\mathcal{D}}}^\beta(\mathcal{A}(S)) - \mathcal{A}(S) \| \big] \leq \vartheta \qquad \text{and} \qquad \mathbb{E}_{S \sim \mathcal{D}^n, \mathcal{A}} \big[ \mathsf{Gap}_{F_{\mathcal{D}}} \big( prox_{F_{\mathcal{D}}}^\beta(\mathcal{A}(S)) \big) \big] \leq \vartheta.$$

In the above, some technical caution must be taken to define the gap function in the stochastic non-smooth case, which we defer to Appendix C.2.3. Although not defined under this name, this is precisely the certificate achieved in weakly-convex SO in recent literature [DG19, DD19].

## 5.2 Proximally Guided Private Stochastic Mirror Descent

Now we provide an algorithm for DP-SO with weakly convex losses that certifies proximal near stationarity (Algorithm 2).

---

**Algorithm 2** Proximally Guided Private Stochastic Mirror Descent

---

**Require:** Private dataset $S = (z_1, \ldots, z_n) \in \mathcal{Z}^n$, number of rounds $R$, $\beta > 0$ regularization parameter
1: Let $\bar{p} = \max\{p, 1 + 1/\log d\}$, and choose initialization $w_1 \in \mathcal{W}$
2: **for** $r = 1$ to $R$ **do**
3:     Extract batch $S_r$ from $S \setminus \bigcup_{l < r} S_r$ of size, $n_r = n/R$
4:     Let $w_{r+1}$ the the output of $\mathcal{A}_{\mathsf{SC}}$ on data $S_r$ for the objective

$$\min_{w \in \mathcal{W}} F_r(w) := \Big\{ F_{\mathcal{D}}(w) + \frac{\beta}{2} \| w - w_r \|_{\bar{p}}^2 \Big\} \tag{6}$$

5: Output: Output $\overline{w}^R$, chosen uniformly at random from $(w_r)_{r \in [R]}$.

---

This algorithm is inspired by the *proximally guided stochastic subgradient method* of Davis and Grimmer [DG19], where the proximal subproblems are solved using an optimal algorithm for DP-SCO in the strongly convex case, proposed in [AFKT21], that we call $\mathcal{A}_{\mathsf{SC}}$ (see Theorem 29 in Appendix C.3 for a precise statement). Our algorithm works in rounds $r = 1, \ldots, R$, and at each round the proximal-type mapping subproblem

$$\min_{w \in \mathcal{W}} F_r(w) = \Big\{ F_{\mathcal{D}}(w) + \frac{\beta}{2} \| w - w_r \|_{\bar{p}}^2 \Big\},$$

is approximately solved using a separate minibatch of size $n/R$ with algorithm $\mathcal{A}_{\mathrm{SC}}$. The $\bar{p}$ used in the subproblem norm is chosen as $\bar{p} = \max\{p, 1 + 1/\log d\}$, in order to control the strong convexity. Finally, the output is chosen uniformly at random from the iterates.

Below, we formally state our main result in this section. The proof can be found in Appendix C.3.

**Theorem 16.** *Consider the $\ell_p$ setting of $\rho$-weakly convex stochastic optimization, where $1 \le p \le 2$. Let $\kappa = \min\{1/(p-1), \log d\}$, $\tilde{\kappa} = 1 + \log d \cdot \mathbf{1}(p < 2)$, and $\beta = 2\rho\kappa$. Suppose that $nd \ge \rho D/L_0$. Then the output of the Proximally Guided Private Stochastic Mirror Descent (Algorithm 2 in Appendix C.3) is $(\varepsilon, \delta)$-DP, and for $R = \left\lfloor \min\left\{ \sqrt{\frac{nD\rho}{\kappa L_0}}, \frac{1}{(\tilde{\kappa}\kappa^2)^{1/3}}\left(\frac{D(n\varepsilon)^2\rho}{L_0 d \log(1/\delta)}\right)^{1/3} \right\} \right\rfloor$, it is guaranteed to provide a $(\vartheta, \beta)$-proximal nearly stationary point, with*

$$\vartheta = \frac{\max\{1, 2\rho D\kappa\}}{\sqrt{\rho}} O\left(\frac{L_0^{3/4}(\kappa D)^{1/4}}{[n\rho]^{1/4}} + (\tilde{\kappa}\kappa^2)^{1/6}(L_0^2 D)^{1/3}\left(\frac{d\log(1/\delta)}{(n\varepsilon)^2\rho}\right)^{1/6}\right). \tag{7}$$

*The running time of this algorithm is upper bounded by $O\left(\log n \cdot \log\log n \cdot \min\left(n^{3/2}\sqrt{\log d}, n^2\varepsilon/\sqrt{d}\right)\right).$*

*Remark* 17. Some comments are in order. First, the bound from eqn. (7) takes the particular form for $p = 1$ and $p = 2$, respectively,

$$\vartheta = \begin{cases} \frac{\max\{1, 2\rho D\log d\}}{\sqrt{\rho}} O\left(\frac{L_0^{3/4}(D\log d)^{1/4}}{[n\rho]^{1/4}} + \sqrt{\log d}(L_0^2 D)^{1/3}\left(\frac{d\log(1/\delta)}{(n\varepsilon)^2\rho}\right)^{1/6}\right) & p = 1 \\[2ex] \frac{\max\{1, 2\rho D\}}{\sqrt{\rho}} O\left(\frac{L_0^{3/4}(D)^{1/4}}{[n\rho]^{1/4}} + (L_0^2 D)^{1/3}\left(\frac{d\log(1/\delta)}{(n\varepsilon)^2\rho}\right)^{1/6}\right) & p = 2. \end{cases}$$

Second, the upper bound in running time can be further refined, taking into account the precise value of $R$. We omit the resulting bound, only for simplicity. Third, we note that the accuracy of our algorithm can be further refined, if one considers the initial optimality gap, $\Delta_F = F_{\mathcal{D}}(w_1) - F_{\mathcal{D}}(w^*)$, instead of the crude upper bound $\Delta_F \le L_0 D$. We make this choice only for simplicity, and to keep consistency with the previous sections. Finally, note that the algorithm $\mathcal{A}_{\mathrm{SC}}$ requires performing Bregman projections, as it uses stochastic mirror-descent as subroutine.

## Acknowledgements

RB's and MM's research is supported by NSF Award AF-1908281, Google Faculty Research Award, and the OSU faculty start-up support. CG's research is partially supported by INRIA through the INRIA Associate Teams project and FONDECYT 1210362 project.

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
