# A  Missing Details of Section 3

## A.1  Missing Details of Section 3.1

### A.1.1  Moreau Envelope Smoothing

Let $\mathcal{M}$ be a (potentially unbounded) closed interval, $y \in \mathbb{R}$, and $\beta > 0$. Consider a function $\ell^{(y)} : \mathcal{M} \mapsto \mathbb{R}$ as in Definition 2. The $\beta$-Moreau envelope of $\ell^{(y)}$ is given as

$$\ell_\beta^{(y)}(m) := \min_{u \in \mathcal{M}} \left[ \ell^{(y)}(u) + \frac{\beta}{2} |u - m|^2 \right].$$

Denote the proximal operator with respect to $\ell^{(y)}$ as

$$\mathrm{prox}_{\ell^{(y)}}^\beta(m) = \arg \min_{u \in \mathcal{M}} \left[ \ell^{(y)}(u) + \frac{\beta}{2} |u - m|^2 \right].$$

For convex functions, the Moreau envelope satisfies the following properties.

**Lemma 18.** *(See [Nes05, Can11]) Let $\ell^{(y)} : \mathcal{M} \mapsto \mathbb{R}$ be a convex function and $L_0$-Lipschitz. Then the following hold:*

(a) $\ell_\beta^{(y)}$ *is convex, $2L_0$-Lipschitz and $\beta$-smooth.*

(b) $\ell_\beta^{(y)\prime}(m) = \beta[m - \mathrm{prox}_{\ell^{(y)}}^\beta(m)].$

(c) $\ell_\beta^{(y)}(m) \leq \ell^{(y)}(m) \leq \ell_\beta^{(y)}(m) + L_0^2/(2\beta)$

### A.1.2  Proof of lemma 3

The Lipschitzness guarantee follows straightforwardly from Lemma 18. For the smoothness guarantee, note that $\nabla f_\beta(w, (x, y)) = \ell_\beta^{(y)\prime}(\langle w, x \rangle) x$. Since $\ell_\beta^{(y)}$ is $\beta$-smooth, for any $w, w' \in \mathcal{W}$ we have

$$\begin{aligned}
\|\nabla f_\beta(w, (x, y)) - \nabla f_{z,\beta}(w', (x, y))\|_* &= \|\ell_\beta^{(y)\prime}(\langle w, x \rangle)x - \ell_\beta^{(y)\prime}(\langle w', x \rangle)x\|_* \\
&= \|x\|_* \cdot |\ell_\beta^{(y)\prime}(\langle w, x \rangle) - \ell_\beta^{(y)\prime}(\langle w', x \rangle)| \\
&\leq \|x\|_* \beta |\langle w, x \rangle - \langle w', x \rangle| \\
&\leq \|x\|_*^2 \beta \|w - w'\|,
\end{aligned}$$

where the last step follows from the definition of the dual norm. For the accuracy, by the guarantees of the Moreau envelope of $\ell^{(y)}$ it holds that for all $w \in \mathbb{R}^d$ and $(x, y) \in \mathcal{X} \times \mathbb{R}$ that

$$|f(w, (x, y)) - f_\beta(w, (x, y))| = |\ell^{(y)}(\langle w, x \rangle) - \ell_\beta^{(y)}(\langle w, x \rangle)|$$

$$\leq \frac{L_0^2}{2\beta}.$$

### A.1.3  Proof of Lemma 4

Let $x, y$ and $w$ be the inputs to Algorithm 3. Note as defined in Algorithm 3, $m = \langle w, x \rangle$ and $\mathcal{P} = \mathcal{M} \cap \left[ m - \frac{2L_0}{\beta}, m + \frac{2L_0}{\beta} \right]$. Define $h_\beta(u) \triangleq \ell^{(y)}(u) + \frac{\beta}{2} |u - m|^2$, i.e. the proximal loss. Let $u^* = \arg \min_{u \in \mathbb{R}} \{h_\beta(u)\}$. We first show that $|\bar{u} - u^*|$ is small by noting that lines 1-10 of Algorithm 3 implement the bisection method on $h_\beta$. Thus, so long as $\mathcal{P}$ is a closed interval, $u^* \in \mathcal{P}$,

---

**Algorithm 3** $\mathcal{O}_{\beta,\alpha,R}$: Gradient Oracle for Smoothed GLL

---

**Require:** Parameter Vector $w \in \mathcal{W}$, Datapoint $(x,y) \in (\mathcal{X} \times \mathbb{R})$

  1: $m = \langle w, x \rangle$

  2: Let $[a,b] = \mathcal{M} \cap \left[ m - \frac{2L_0}{\beta}, m + \frac{2L_0}{\beta} \right]$

  3: $T = \left\lceil \log_2\left( \frac{16L_0^2 R^2}{\alpha^2} \right) \right\rceil$

  4: **for** $t = 1$ to $T$ **do**

  5:     Let $m_t = \frac{a+b}{2}$

  6:     **if** $\ell^{(y)}\left(\frac{a+m_t}{2}\right) + |\frac{a+m_t}{2} - m|^2 \geq \ell^{(y)}\left(\frac{m_t+b}{2}\right) + |\frac{m_t+b}{2} - m|^2$ **then**

  7:       $b = m_t$

  8:     **else**

  9:       $a = m_t$

10: $\bar{u} = \underset{\{m_t : t \in [T]\}}{\arg\min} \{ \ell^{(y)}(m_t) + |m_t - m|^2 \}$

11: Output: $\beta(m - \bar{u})x$

---

and $\max_{u \in \mathcal{P}} \{ h_\beta(u) - h_\beta(u^*) \} \leq \tau$, standard guarantees of the bisection method give that $h_\beta(\bar{u}) - h_\beta(u^*) \leq \tau 2^{-T}$ (see, e.g., [Nem95, Theorem 1.1.1]). Clearly $\mathcal{P}$ is a closed interval since $\mathcal{M}$ is closed. To see that $u^* \in \mathcal{P}$, note that since $u^*$ is the minimizer of $h_\beta$ it holds that

$$ 0 \leq \ell^{(y)}(m) + \frac{\beta}{2}|m - m|^2 - \ell^{(y)}(u^*) - \frac{\beta}{2}|u^* - m|^2 = \ell^{(y)}(m) - \ell^{(y)}(u^*) - \frac{\beta}{2}|u^* - m|^2. $$

Further since $\ell^{(y)}$ is $L_0$-Lipschitz we have that $\ell^{(y)}(m) - \ell^{(y)}(u^*) \leq L_0|u^* - m|$. Using this fact in the above inequality we obtain $|m - u^*| \leq 2L_0/\beta$ and thus $u^* \in \mathcal{P}$. Using the bound on the radius of $\mathcal{P}$ and Lipschitz constant of $\ell^{(y)}$ it holds that $\tau \leq 8L_0^2/\beta$. The setting of $T = \left\lceil \log_2\left( \frac{16L_0^2 R^2}{\alpha^2} \right) \right\rceil$ and the accuracy gaurantees of the bisection method then gives that $h_\beta(\bar{u}) - h_\beta(u^*) \leq \frac{\alpha^2}{2\beta R^2}$. Since $h_\beta$ is $\beta$-strongly convex we then have

$$ |\bar{u} - u^*| \leq \sqrt{\frac{2\left(h_\beta(\bar{u}) - h_\beta(u^*)\right)}{\beta}} \leq \frac{\alpha}{\beta R}. $$

The accuracy guarantee $\|\mathcal{O}_{\beta,\alpha,R}(w,(x,y)) - \nabla f_\beta(w,(x,y))\|_* \leq \alpha$ then follows straightforwardly using part (b) of Lemma 18 and the facts that $\|x\|_* \leq R$ and $u^* = \mathrm{prox}_{\ell^{(y)}}^\beta(m)$.

### A.2 Proof of Theorem 5

The proof of convergence follows similarly to [FKT20], but additionally we account for the change in gradient sensitivity and extra error introduced by using the approximate gradient oracle of the smoothed loss, $\mathcal{O}_{\beta,\alpha,R}$. Let $\mathrm{PSGD}(\mathcal{O}, \eta, w_0, T)$ (used in Algorithm 4) denote the process which computes $w_t = \Pi_{\mathcal{W}}[w_{t-1} + \eta \mathcal{O}(w_{t-1})] : \forall t \in [T]$, where $\Pi_{\mathcal{W}}$ is the projection onto constraint set $\mathcal{W}$. By Lemma 3, $f_\beta$ is a $(2L_0 R)$-Lipschitz and $(\beta R^2)$-smooth loss function. Further, the increase in error due to using $\alpha$-approximate gradients in PSGD is at most $2\alpha D$ (see, e.g., [FGV17, BFTT19]). Let $F_{\beta,\mathcal{D}}(w) = \underset{z \sim \mathcal{D}}{\mathbb{E}} [f_\beta(w)]$ and let $w_\beta^* = \underset{w \in \mathcal{W}}{\arg\min}\{F_{\beta,\mathcal{D}}(w)\}$. For notational convenience, let $w_0 = w_\beta^*$ and $\xi_0 = \tilde{w}_0 - w_\beta^*$. We have (following [FKT20, Proof of Theorem 4.4]):

**Algorithm 4** Phased SGD for GLL

**Require:** Private dataset $(z_1, \ldots, z_n) \in \mathcal{Z}^n$, constraint set $\mathcal{W} \subseteq \mathbb{R}^d$, privacy parameters $(\varepsilon, \delta)$ s.t. $\varepsilon \le \sqrt{\log(1/\delta)}$, constraint diameter (for constrained case) $D$, Lipschitz constant $L_0$, smoothness parameter $\beta$, oracle accuracy $\alpha$, feature vector norm bound $R$
1: Let $\tilde{w}_0 \in \mathcal{W}$ be arbitrary
2: $\rho = \frac{\varepsilon}{2\sqrt{\log(1/\delta)}}$
3: $K = \log_2(n)$
4: For **Constrained** setting: $\eta = \frac{D}{3L_0 R} \min\{\frac{\rho}{\sqrt{d}}, \frac{1}{\sqrt{n}}\}$
5: For **Unconstrained** setting: $\eta = \frac{1}{3L_0 R} \min\{\frac{\rho}{\sqrt{\theta}}, \frac{1}{\sqrt{n}}\}$, where $\theta$ is an upper bound on the *expected rank* of $\sum_{i=1}^n x_i x_i^\top$. (Note that we always have $\theta \le n$.)
6: $s = 1$
7: **for** $k = 1$ to $K$ **do**
8: $\quad T_k = \frac{n}{2^k}$
9: $\quad \eta_k = \frac{\eta}{4^k}$
10: $\quad$ Initialize PSGD algorithm of [FKT20] (over domain $\mathcal{W}$) at $\tilde{w}_{k-1}$ and run with oracle $\mathcal{O}_{\beta, \alpha, R}$ in place of $\nabla f$ and step size $\eta_k$ for $T_k$ steps over dataset $\{z_s, ..., z_{s+T_k}\}$. Let $w_k$ be the average of the iterate of PSGD.
11: $\quad \tilde{w}_k = w_k + \xi_k$ where $\xi_k \sim \mathcal{N}(0, \mathbb{I}_d \sigma_k^2)$ with $\sigma_k = \frac{4L_0 R \eta_k}{\rho}$
12: $\quad s = s + T_k$
13: Output: $\tilde{w}_K$

$$\mathbb{E}\left[F_{\beta, \mathcal{D}}(\tilde{w}_K) - F_{\beta, \mathcal{D}}(w_\beta^*)\right] = \sum_{k=1}^K \mathbb{E}\left[F_{\beta, \mathcal{D}}(w_k) - F_{\beta, \mathcal{D}}(w_{k-1})\right] + \mathbb{E}\left[F_{\beta, \mathcal{D}}(\tilde{w}_K) - F_{\beta, \mathcal{D}}(w_K)\right]$$

$$\le \sum_{k=1}^K \left(\frac{d\sigma_{k-1}^2}{2\eta_k T_k} + 2\eta_k L_0^2 R^2 + 2D\alpha\right) + 2L_0 R \, \mathbb{E}[\|\xi_K\|_2]$$

$$= \sum_{k=2}^K \left(\frac{d\sigma_{k-1}^2}{2\eta_k T_k} + 2\eta_k L_0^2 R^2\right) + 2L_0 R \sqrt{d} \sigma_K + 2DK\alpha.$$

Where the first inequality follows from the convergence of PSGD [FKT20, Lemma 4.5]. By the setting of $\alpha = \frac{L_0 R}{n \log(n)}$, we have $2DK\alpha = \frac{2L_0 RD}{n}$. It can be verified that the rest of the expression is $O\left(L_0 RD \left(\frac{1}{\sqrt{n}} + \frac{\sqrt{d}}{\rho n}\right)\right)$ (see [FKT20, Proof of Theorem 4.4]). To convert to population loss with respect to the original function, we provide the following analysis. Let $w^* = \min_{w \in \mathcal{W}} F_{\mathcal{D}}(w^*)$. By Lemma 3 we have for any $w \in \mathcal{W}$

$$F_{\mathcal{D}}(w) - F_{\mathcal{D}}(w^*) \le F_{\beta, \mathcal{D}}(w) - F_{\beta, \mathcal{D}}(w^*) + \frac{L_0^2}{\beta}$$

$$\le F_{\beta, \mathcal{D}}(w) - F_{\beta, \mathcal{D}}(w_\beta^*) + \frac{L_0^2}{\beta}.$$

Thus by the setting $\beta = \sqrt{n} L_0/(RD)$ we have

$$\mathbb{E}\left[F_{\mathcal{D}}(\tilde{w}_K) - F_{\mathcal{D}}(w^*)\right] = O\left(L_0 RD \left(\frac{1}{\sqrt{n}} + \frac{\sqrt{d}}{\rho}\right)\right).$$

Plugging in our value of $\rho$ into the above we have the final result.

$$\mathbb{E}\left[F_{\mathcal{D}}(\tilde{w}_K) - F_{\mathcal{D}}(w^*)\right] = O\left(L_0 RD\left(\frac{1}{\sqrt{n}} + \frac{\sqrt{d\log(1/\delta)}}{n\varepsilon}\right)\right).$$

For privacy, note that $\|\mathcal{O}_{\beta,\alpha,R}(w,z)\| \leq (2L_0 R + \frac{L_0 R}{n})$, and thus the sensitivity of the approximate gradient is bounded by $3L_0 R$. Thus, by setting the parameters of Phased SGD as they would be for a $(3L_0 R)$-Lipschitz function, Lemma 4.5 of [FKT20] implies that Algorithm 4 satisfies $(\varepsilon, \delta)$-DP so long as $\eta \leq \frac{2}{\beta R^2}$. It's easy to see that the condition on $\eta$ holds.

**Proof for the Unconstrained Case:** We show here a detailed proof for the excess risk guarantees for our construction for (possibly non-smooth) GLLs in the *unconstrained* case (Algorithm 4). This result was mentioned in the remark after Theorem 5. Unlike the construction in [SSTT21] with super-linear time, the construction we give here runs in *near-linear time*.

Before presenting our result for this setting, a few preliminaries are necessary. Let $V$ be a matrix whose columns are an eigenbasis for $\sum_{i=1}^{n} x_i x_i^\top$. For any $u, u' \in \mathbb{R}^d$, let $\|u\|_V = \sqrt{u^\top V V^\top u}$ denote the semi-norm of $u$ induced by $V$, and let $\langle u, u'\rangle_V = u^\top V V^T u'$. Here, we assume knowledge of some upper bound $\theta$ on $\mathbb{E}_{S\sim\mathcal{D}}[\mathsf{Rank}(V)]$. Note that this is no loss of generality since we always have $\mathbb{E}_{S\sim\mathcal{D}}[\mathsf{Rank}(V)] \leq n$; hence, if we don't have this additional knowledge, we can set $\theta = n$.

**Theorem 19.** *Let $\mathcal{W} = \mathbb{R}^d$. Let $f : \mathcal{W} \times (\mathcal{X} \times \mathbb{R}) \to \mathbb{R}$ be a $L_0$-Lipschitz and $R$-bounded GLL with respect to $\|\cdot\|_2$. Let $\beta = \sqrt{n}L_0/R$, $\alpha = \frac{L_0 R}{n\log n}$. Then Phased-SGD run with oracle $\mathcal{O}_{\beta,\alpha,R}$ and dataset $S \in \mathcal{Z}^n$ satisfies $(\varepsilon, \delta)$ differential privacy and has running time $O(n\log n)$. Further, if $S \sim \mathcal{D}^n$ the output of Phased-SGD has expected excess population risk*

$$O\left(L_0 R\left(\|\tilde{w}_0 - w_\beta^*\|^2 + 1\right)\left(\frac{\sqrt{\theta\log(1/\delta)}}{n\varepsilon} + \frac{1}{\sqrt{n}}\right)\right)$$

To prove the claim, we start by providing the following lemma. As before, denote $w_0 = w_\beta^*$ and $\xi_0 = \tilde{w}_0 - w_\beta^*$.

**Lemma 20.** *Let $\alpha, \beta, R$ be as in Theorem 19. Then the output, $w_k$, of phase $k$ of Phased SGD using $\mathcal{O}_{\beta,\alpha,R}$ satisfies*

$$\mathbb{E}\left[F_{\beta,D}(w_k) - F_{\beta,D}(w_{k-1})\right] \leq \frac{\mathbb{E}\left[\|\tilde{w}_{k-1} - w_{k-1}\|_V^2\right]}{2\eta_k T_k} + \frac{5\eta_k L_0^2 R^2}{2} + \frac{L_0 R\left(\mathbb{E}\left[\|\tilde{w}_{k-1} - w_{k-1}\|_V\right] + 1\right)}{\sqrt{n}\log(n)}.$$

*Proof.* Let $\{u_0, \ldots, u_{T_k}\}$ denote the iterates generated by round $k$ of PSGD (where $u_0 = \tilde{w}_{k-1}$), and let $z_t$ be the datapoint sampled during iteration $t$. For all $t \in \{0, \ldots T_k\}$, define the potential function $\Phi^{(t)} \triangleq \|u_t - w_{k-1}\|_V^2$. Using standard algebraic manipulation, we have

$$\Phi^{(t+1)} = \Phi^{(t)} - 2\eta_k\langle\mathcal{O}_{\beta,\alpha,R}(u_t, z_t), u_t - w_{k-1}\rangle_V + \eta_k^2\|\mathcal{O}_{\beta,\alpha,R}(u_t, z_t)\|_V^2$$
$$\leq \Phi^{(t)} - 2\eta_k\langle\nabla f_\beta(u_t, z_t), u_t - w_{k-1}\rangle_V + 2\eta_k\alpha\|u_t - w_{k-1}\|_V + \eta_k^2(\alpha^2 + 4L_0^2 R^2),$$

where the inequality follows from the fact that $\|\mathcal{O}_{\beta,\alpha,R}(u_t, z_t) - \nabla f_\beta(u_t, z_t)\| \leq \alpha$ and the nonexpansiveness of the projection onto the span of $V$. Since the gradient is in the span of $V$, we have

$$\Phi^{(t+1)} \leq \Phi^{(t)} - 2\eta_k\langle\nabla f_\beta(u_t, z_t), u_t - w_{k-1}\rangle + 2\eta_k\alpha\|u_t - w_{k-1}\|_V + \eta_k^2(\alpha^2 + 4L_0^2 R^2).$$

Hence

$$\langle\nabla f_\beta(u_t, z_t), u_t - w_{k-1}\rangle \leq \frac{\Phi^{(t)} - \Phi^{(t+1)}}{2\eta_k} + \alpha\|u_t - w_{k-1}\|_V + \frac{\eta_k}{2}(\alpha^2 + 4L_0^2 R^2).$$

Taking the expectation w.r.t. all randomness (i.e., w.r.t. $S \sim \mathcal{D}^n$ and the Gaussian noise random variables), we have

$$\mathbb{E}\left[\langle\nabla F_{\beta,D}(u_t), u_t - w_{k-1}\rangle\right] \leq \frac{\mathbb{E}\left[\Phi^{(t)} - \Phi^{(t+1)}\right]}{2\eta_k} + \alpha\mathbb{E}\left[\|u_t - w_{k-1}\|_V\right] + \frac{\eta_k}{2}(\alpha^2 + 4L_0^2 R^2).$$

Moreover, by the convexity of $F_{\beta,D}$ we have $\mathbb{E}\left[\langle \nabla F_{\beta,D}(u_t), u_t - w_{k-1}\rangle\right] \geq \mathbb{E}\left[F_{\beta,D}(u_t) - F_{\beta,D}(w_{k-1})\right]$. Combining this inequality with the above, and using the fact that $w_k = \frac{1}{T_k}\sum_{t=1}^{T_k} u_t$ together with the convexity of $F_{\beta,D}$, we have

$$
\mathbb{E}\left[F_{\beta,D}(w_k) - F_{\beta,D}(w_{k-1})\right] \leq \frac{1}{T_k}\sum_{t=1}^{T_k}\Big(\mathbb{E}\left[F_{\beta,D}(u_t) - F_{\beta,D}(w_{k-1})\right]\Big)
$$

$$
\leq \frac{\mathbb{E}\left[\Phi^{(0)}\right]}{2\eta_k T_k} + \frac{\alpha}{T_k}\mathbb{E}\left[\sum_{t=1}^{T_k}\|u_t - w_{k-1}\|_V\right] + \frac{\eta_k}{2}(\alpha^2 + 4L_0^2 R^2).
$$

To bound $\mathbb{E}\left[\sum_{t=1}^{T_k}\|u_t - w_{k-1}\|_V\right]$ in the above, observe that,

$$
\|u_t - w_{k-1}\|_V \leq \|u_{t-1} - w_{k-1}\|_V + \|u_t - u_{t-1}\|_V
$$

$$
\vdots
$$

$$
\leq \|\tilde{w}_{k-1} - w_{k-1}\|_V + \sum_{j=1}^{t}\|u_j - u_{j-1}\|_V.
$$

Hence

$$
\mathbb{E}\left[\|u_t - w_{k-1}\|_V\right] \leq \mathbb{E}\left[\|\tilde{w}_{k-1} - w_{k-1}\|_V\right] + \sum_{j=1}^{t}\mathbb{E}\left[\|u_j - u_{j-1}\|_V\right]
$$

$$
\leq \mathbb{E}\left[\sqrt{\Phi^{(0)}}\right] + \eta_k t(2L_0 R + \alpha),
$$

where the last inequality follows from the definition of $\Phi^{(0)}$ and the fact that $\mathbb{E}\left[\|u_j - u_{j-1}\|_V\right] = \eta_k \mathbb{E}\left[\|\mathcal{O}_{\beta,\alpha,R}(u_{j-1}, z_{j-1})\|\right] \leq \eta_k(2L_0 R + \alpha)$. Thus we have

$$
\mathbb{E}\left[F_{\beta,D}(w_k) - F_{\beta,D}(w_{k-1})\right] \leq \frac{\mathbb{E}\left[\Phi^{(0)}\right]}{2\eta_k T_k} + \alpha\left(\mathbb{E}\left[\sqrt{\Phi^{(0)}}\right] + T_k \eta_k(2L_0 R + \alpha)\right) + \frac{\eta_k}{2}(\alpha^2 + 4L_0^2 R^2)
$$

$$
= \frac{\mathbb{E}\left[\Phi^{(0)}\right]}{2\eta_k T_k} + \frac{5\eta_k L_0^2 R^2}{2} + \alpha\left(\mathbb{E}\left[\sqrt{\Phi^{(0)}}\right] + 3T_k \eta_k L_0 R\right).
$$

The last step follows from the fact that $\alpha = \frac{L_0 R}{n\log(n)} \leq L_0 R$. Further, since $\eta_k = \frac{1}{3L_0 R_0}\min\{\frac{\rho}{\sqrt{\theta}}, \frac{1}{\sqrt{n}}\} \leq \frac{1}{3L_0 R\sqrt{n}}$ and $T_k \leq n$ it holds that $3T_k \eta_k L_0 R \leq \sqrt{n}$. Thus by the setting of $\alpha$, we have

$$
\mathbb{E}\left[F_{\beta,D}(w_k) - F_{\beta,D}(w_{k-1})\right] \leq \frac{\mathbb{E}\left[\Phi^{(0)}\right]}{2\eta_k T_k} + \frac{5\eta_k L_0^2 R^2}{2} + \frac{L_0 R\left(\mathbb{E}\left[\sqrt{\Phi^{(0)}}\right] + 1\right)}{\sqrt{n}\log(n)}.
$$

$\square$

Recall that we denote $w_0 = w_\beta^*$ and $\xi_0 = \tilde{w}_0 - w_\beta^*$. Using the above lemma and noting that $\tilde{w}_{k-1} - w_{k-1} = \xi_{k-1}$, the excess risk of the $\tilde{w}_K$ is bounded by

$$
\begin{aligned}
\mathbb{E}\left[F_{\beta,\mathcal{D}}(\tilde{w}_K) - F_{\beta,\mathcal{D}}(w_\beta^*)\right] &= \sum_{k=1}^{K} \mathbb{E}\left[F_{\beta,\mathcal{D}}(w_k) - F_{\beta,\mathcal{D}}(w_{k-1})\right] + \mathbb{E}\left[F_{\beta,\mathcal{D}}(\tilde{w}_K) - F_{\beta,\mathcal{D}}(w_K)\right] \\
&\leq \sum_{k=1}^{K} \left(\frac{\mathbb{E}\left[\|\xi_{k-1}\|_V^2\right]}{2\eta_k T_k} + \frac{5\eta_k L_0^2 R^2}{2} + \frac{L_0 R\left(\mathbb{E}\left[\|\xi_{k-1}\|_V\right] + 1\right)}{\sqrt{n}\log(n)}\right) \\
&\quad + \mathbb{E}\left[F_{\beta,\mathcal{D}}(\tilde{w}_K) - F_{\beta,\mathcal{D}}(w_K)\right]. \quad\quad (8)
\end{aligned}
$$

Note that for any $2 \leq k \leq K$, we have

$$
\mathbb{E}\left[\|\xi_{k-1}\|_V^2\right] = \mathbb{E}_V\left[\mathbb{E}_{\xi_{k-1}}\left[\xi_{k-1}^\top V V^\top \xi_{k-1} | V\right]\right] \leq \mathbb{E}_V\left[\mathsf{Rank}(V)\right]\sigma_{k-1}^2 \leq \theta\sigma_{k-1}^2
$$

At round $k = 1$, we simply have $\mathbb{E}_V\left[\|\xi_0\|_V\right] \leq \|\tilde{w}_0 - w_\beta^*\|$. Finally, since $f$ is a GLL, the expected increase in loss due to $\xi_K$ is bounded as

$$
\begin{aligned}
\mathbb{E}\left[F_{\beta,D}(\tilde{w}_K) - F_{\beta,D}(w_K)\right] &= \mathbb{E}_{(x,y)\sim\mathcal{D}}\left[\mathbb{E}_{\xi_K}\left[\ell_\beta^{(y)}(\langle\tilde{w}_K, x\rangle) - \ell_\beta^{(y)}(\langle w_K, x\rangle)\right]\right] \\
&\leq \mathbb{E}_{(x,y)\sim\mathcal{D}}\left[\mathbb{E}_{\xi_K}\left[L_0|\langle\xi_K, x\rangle|\right]\right] \\
&\leq L_0 R\sigma_K \\
&= \frac{L_0 R}{4^{K-1}\sqrt{n}} \\
&= \frac{L_0 R}{4n^{5/2}}
\end{aligned}
$$

The second inequality follows from the fact that $\ell_\beta^{(y)}$ is $L_0$-Lipschitz, and the last two steps follow form the fact that $\sigma_k \leq \frac{1}{4^{k-1}\sqrt{n}}$ and $K = \log_2(n)$. Thus, using inequality (8) above, we have

$$
\begin{aligned}
\mathbb{E}\left[F_{\beta,\mathcal{D}}(\tilde{w}_K) - F_{\beta,\mathcal{D}}(w_\beta^*)\right] &= O\left(L_0 R\left(\|\tilde{w}_0 - w_\beta^*\|^2 + 1\right)\left(\frac{\sqrt{\theta}}{n\rho} + \frac{1}{\sqrt{n}}\right)\right) + \sum_{k=2}^{K}\left(\frac{\theta\sigma_{k-1}^2}{2\eta_k T_k} + \frac{5\eta_k L_0^2 R^2}{2} + \frac{L_0 R(\sqrt{\theta}\sigma_{k-1} + 1)}{\sqrt{n}\log(n)}\right) \\
&\quad + \frac{L_0 R}{4n^{5/2}} \\
&= O\left(L_0 R\left(\|\tilde{w}_0 - w_\beta^*\|^2 + 1\right)\left(\frac{\sqrt{\theta}}{n\rho} + \frac{1}{\sqrt{n}}\right)\right) + \sum_{k=2}^{K}\left(\frac{\theta\sigma_{k-1}^2}{2\eta_k T_k} + \frac{5\eta_k L_0^2 R^2}{2}\right) + \frac{3L_0 R}{\sqrt{n}} \\
&= O\left(L_0 R\left(\|\tilde{w}_0 - w_\beta^*\|^2 + 1\right)\left(\frac{\sqrt{\theta}}{n\rho} + \frac{1}{\sqrt{n}}\right)\right) + O\left(L_0 R\left(\frac{\sqrt{\theta}}{n\rho} + \frac{1}{\sqrt{n}}\right)\right) \\
&= O\left(L_0 R\left(\|\tilde{w}_0 - w_\beta^*\|^2 + 1\right)\left(\frac{\sqrt{\theta}}{n\rho} + \frac{1}{\sqrt{n}}\right)\right).
\end{aligned}
$$

The first line comes from bounding the term corresponding to $k = 1$ in the sum in (8), and the settings of $\eta_1 = \frac{\rho}{12 L_0 R\sqrt{n}}$ and $T_1 = n/2$. The second equality follows from the fact that $\sqrt{\theta}\sigma_{k-1} = 4\sqrt{\theta}L_0 R\eta_{k-1}/\rho \leq 4\sqrt{\theta}L_0 R\eta/\rho \leq 2$, and the

fact that $K = \log_2(n)$. The third step follows from the choices of $\eta_k, T_k$ and $\sigma_{k-1}$. To reach the final result, we convert the guarantee above to a guarantee for the original (unsmoothed) loss and use the setting of $\beta = \sqrt{n}L_0/R$ (as done in the proof of Theorem 5).

## A.3   Proof of Theorem 6

---

**Algorithm 5** Noisy Frank Wolfe

---

**Require:** Private dataset $S = (z_1, ..., z_n) \in \mathcal{Z}^n$, polyhedral set $\mathcal{W}$ with vertices $\mathcal{V}$, Lipschitz constant $L_0$, constraint diameter $D$, privacy parameters $(\varepsilon, \delta)$, smoothness parameter $\beta$, oracle accuracy $\alpha$, feature vector norm bound $R$

1: Let $w_1 \in \mathcal{W}$ be arbitrary
2: $T = \frac{n\varepsilon}{\log(|V|)\log(n)\sqrt{\log(1/\delta)}}$
3: $s = \frac{3L_0 R D \sqrt{8T\log(1/\delta)}}{n\varepsilon}$
4: **for** $t = 1$ to $T$ **do**
5: $\quad \tilde{\nabla}_t = \frac{1}{n}\sum_{z \in S} \mathcal{O}_{\beta,\alpha,R}(w_t, z)$
6: $\quad$ Draw $\{b_{v,t}\}_{v \in \mathcal{V}}$ i.i.d from $\mathsf{Lap}(s)$
7: $\quad \tilde{v}_t = \arg\min_{v \in \mathcal{V}}\{\langle v, \tilde{\nabla}_t \rangle + b_{v,t}\}$
8: $\quad w_{t+1} = (1 - \mu_t)w_t + \mu_t \tilde{v}_t$, where $\mu_t = \frac{3}{t+2}$
9: Output: $w_T$

---

The proof follows from the analysis of noisy Frank Wolfe from [TTZ16]. Let $F_{\beta,S}(w) = \frac{1}{n}\sum_{z \in S} f_\beta(w, z)$. Define $w_{\beta,S}^*$ as the minimizer $F_{\beta,S}$ in $\mathcal{W}$.

Define $\gamma_t = \langle \tilde{v}_t, \tilde{\nabla}_t \rangle - \min_{v \in \mathcal{V}} \langle v, \tilde{\nabla}_t \rangle$. Since $F_{\beta,S}$ is $(\beta R^2)$-smooth (by Lemma 3), standard analysis of the Noisy Frank-Wolfe algorithm yields (see, e.g., [TTZ15])

$$\mathbb{E}\left[F_{\beta,S}(w_T) - F_{\beta,S}(w_{\beta,S}^*)\right] \leq O\left(\frac{\beta R^2 D^2}{T}\right) + D\sum_{t=1}^{T} \mu_t \mathbb{E}\left[\left\|\tilde{\nabla}_t - \nabla F_{\beta,S}(w_t)\right\|_\infty\right] + \sum_{t=1}^{T} \mu_t \mathbb{E}\left[\gamma_t\right].$$

By a standard argument concerning the maximum of a collection of Laplace random variables, we have for all $t \in [T]$ $\mathbb{E}[\gamma_t] \leq 2s\log(|\mathcal{V}|)$. Note also that for all $t$, by the approximation guarantee of $\mathcal{O}_{\beta,\alpha,R}$, we have (with probability 1) $\left\|\tilde{\nabla}_t - \nabla F_{\beta,S}(w_t)\right\|_\infty \leq \alpha$. Hence,

$$\mathbb{E}\left[F_{\beta,S}(w_T) - F_{\beta,S}(w_{\beta,S}^*)\right] \leq O\left(\frac{\beta R^2 D^2}{T}\right) + \log(T)\left(D\alpha + s\log(|\mathcal{V}|)\right)$$

$$= O\left(\frac{\beta R^2 D^2}{T}\right) + \log(T)\left(\frac{L_0 R D}{n\log(n)} + \frac{L_0 R D \sqrt{8T\log(1/\delta)}\log(|\mathcal{V}|)}{n\varepsilon}\right),$$

where the second equality follows from the setting of $\alpha = \frac{L_0 R}{n\log(n)}$ and the noise parameter $s$.

Using the same argument as in the proof of Lemma 5, we arrive at the following bound on the excess empirical risk for the unsmoothed empirical loss $F_S$:

$$\mathbb{E}[F_S(w_T) - F_S(w_S^*)] = O\left(\frac{\beta R^2 D^2}{T} + \frac{L_0 R D \sqrt{72T\log(1/\delta)}\log(|\mathcal{V}|)\log(T)}{n\varepsilon} + \frac{L_0 R D \log(T)}{n\log(n)} + \frac{L_0^2}{\beta}\right).$$

By the setting of $\beta = \frac{L_0\sqrt{n\varepsilon}}{RD\log^{1/4}(1/\delta)\sqrt{\log(|\mathcal{V}|)\log(n)}}$ and $T = \frac{n\varepsilon}{\log(|\mathcal{V}|)\log(n)\sqrt{\log(1/\delta)}}$,

$$\mathbb{E}\left[F_S(w_T) - F_S(w_S^*)\right] = O\left(\frac{L_0RD\log^{1/4}(1/\delta)\sqrt{\log(|\mathcal{V}|)\log(n)}}{\sqrt{n\varepsilon}}\right).$$

Via a standard Rademacher-complexity argument, we know that the generalization error of GLLs is bounded as $O\left(\frac{L_0RD\sqrt{\log d}}{\sqrt{n}}\right)$ (see [SSBD14] Theorem 26.15). This gives the claimed bound.

The privacy guarantee follows almost the same argument as in [TTZ15]. Note that the sensitivity of the approximate gradients generated by $\mathcal{O}_{\beta,\alpha,R}$ is at most $\frac{3L_0R}{n}$ since $f_\beta$ is $(2L_0R)$-Lipschitz and the error due to the approximate oracle is less than $L_0R$. We then guarantee privacy via a straightforward application of the Report-Noisy-Max algorithm [DR14, BLST10] and advanced composition for differential privacy.

# B   Missing Details of Section 4

**Regularity of Normed Spaces.**   The algorithms we consider in Section 4 can be applied to general spaces whose dual has a sufficiently smooth norm. To quantify this property, we use the notion of *regular spaces* ([JN08]). Given $\kappa \geq 1$, we say a normed space $(\mathbf{E}, \|\cdot\|)$ is $\kappa$-regular, if there exists $1 \leq \kappa_+ \leq \kappa$ and a norm $\|\cdot\|_+$ such that $(\mathbf{E}, \|\cdot\|_+)$ is $\kappa_+$-smooth, i.e.,

$$\|x+y\|_+^2 \leq \|x\|_+^2 + \langle\nabla(\|\cdot\|_+^2)(x), y\rangle + \kappa_+\|y\|_+^2 \qquad (\forall x, y \in \mathbf{E}), \tag{9}$$

and $\|\cdot\|$ and $\|\cdot\|_+$ are equivalent with constant $\sqrt{\kappa/\kappa_+}$:

$$\|x\|^2 \leq \|x\|_+^2 \leq \frac{\kappa}{\kappa_+}\|x\|^2 \qquad (\forall x \in \mathbf{E}). \tag{10}$$

One relevant fact is that $d$-dimensional $\ell_q$ spaces, $2 \leq q \leq \infty$, are $\kappa$-regular with $\kappa = \min(q-1, 2\log d)$. Also, if $\|\cdot\|$ is a polyhedral norm defined over a space $\mathbf{E}$ with unit ball $\mathcal{B}_{\|\cdot\|} = \mathrm{conv}(\mathcal{V})$, then its dual $(\mathbf{E}, \|\cdot\|_*)$ is $(2\log|\mathcal{V}|)$-regular.

**Remark concerning the choice of parameters $R$ and $b$:**   Note that the total number of samples used by our algorithms in Section 4 is $\sum_{r=0}^{R-1}\sum_{t=0}^{2^r-1} b/(t+1) \leq b\sum_{r=0}^{R}(\ln(2^r)+1) = b\sum_{r=0}^{R}(r\ln(2)+1) < bR^2$. Moreover, the batch drawn in each iteration $(r, t)$ is $b/(t+1)$. Hence, for the algorithms to be properly defined, it suffices to have $bR^2 \leq n$ and $b \geq 2^R$. Note that our choices of $R$ and $b$ in both algorithms satisfy these conditions. Note also that we assume w.l.o.g. that $n$ is large enough so that the claimed bounds on the stationarity gap are non-trivial. Hence, the choice of $R$ in each of our algorithms is meaningful.

## B.1   Missing Proofs of Section 4.1

### B.1.1   Proof of Theorem 7

Since the batches used in different rounds $r = 0, \ldots, R-1$ are disjoint, it suffices to prove the privacy guarantee for a given round $r$. The rest of the proof follows by parallel composition of differential privacy. For notational brevity, let $g_r^t = \frac{t+1}{b}\sum_{z\in B_r^t}\nabla f(w_r^t, z)$. By unravelling the recursion in the gradient estimator (Step 11 of Algorithm 1) and using the setting of $\eta_{r,t} = \frac{1}{\sqrt{t+1}}$, we have for any $t \in [2^r - 1]$:

$$\nabla_r^t = a_t^{(1)}\cdot\nabla_r^0 + \sum_{k=1}^{t}\left(a_t^{(k)}\cdot\Delta_r^k + c_t^{(k)}\cdot g_r^k\right) \tag{11}$$

where, for all $k \in [t]$, $a_t^{(k)} = \prod_{j=k}^{t}(1 - \frac{1}{\sqrt{j+1}})$ and $c_t^{(k)} = \frac{1}{\sqrt{k+1}}\prod_{j=k+1}^{t}(1 - \frac{1}{\sqrt{j+1}})$. Note also that $a_t^{(k)} < 1$ and $c_t^{(k)} < 1$ for all $t, k$.

Let $S, S'$ be any neighboring datasets (i.e., differing in exactly one data point). Let $\nabla_r^t, \left\{\Delta_r^k : k \in [t]\right\}, \left\{g_r^k : k \in [t]\right\}$ be the quantities above when the input dataset is $S$; and let $\nabla_r'^t, \left\{\Delta_r'^k, : k \in [t]\right\}, \left\{g_r'^k : k \in [t]\right\}$ be the corresponding quantities when the input dataset is $S'$. Now, since the batches $B_r^0, \ldots, B_r^t$ are disjoint, changing one data point in the input dataset can affect at most one term in the sum (11) above, i.e., it affects either the $\nabla_r^0$ term, or exactly one term corresponding to some $k \in [t]$ in the sum on the right-hand side. Moreover, since $f$ is $L_0$-Lipschitz, we have $\left\|\nabla_r^0 - \nabla_r'^0\right\|_* \leq L_0/b$, and $\left\|g_r^t - g_r'^t\right\|_* \leq L_0(t+1)/b$. Also, by the $L_1$-smoothness of $f$ and the form of the update rule (Step 13 of Algorithm 1), for any $k \in \{1, \ldots, 2^r - 1\}$, we have $\left\|\nabla f(w_r^k, z) - \nabla f(w_r^{k-1}, z)\right\|_* \leq L_1 \left\|w_r^k - w_r^{k-1}\right\| \leq L_1 D\eta_{r,k} \leq L_1 D/\sqrt{k+1}$. Hence, $\left\|\Delta_r^k - \Delta_r'^k\right\|_* \leq \frac{k+1}{b}\frac{L_1 D}{\sqrt{k+1}} = L_1 D\sqrt{k+1}/b$. Using these facts, it is then easy to see that for any $t \in [2^r - 1]$,

$$\left\|\nabla_r^t - \nabla_r'^t\right\|_* \leq \max\left(\frac{L_0}{b}, \frac{(L_0 + L_1 D)\sqrt{t+1}}{b}\right) \leq \frac{(L_0 + L_1 D)2^{r/2}}{b}.$$

Hence, for each $v \in \mathcal{V}$, the global sensitivity of $\langle v, \nabla_r^t\rangle$ is upper bounded by $\frac{D(L_0 + L_1 D)2^{r/2}}{b}$. By the privacy guarantee of the Report Noisy Max mechanism [DR14, BLST10], the setting of the Laplace noise parameter $s_r$ ensures that each iteration $t \in \{0, \ldots, 2^r - 1\}$ is $\frac{\varepsilon 2^{-r/2}}{\sqrt{\log(1/\delta)}}$-DP. Thus, by advanced composition (Lemma 1) applied to the $2^r$ iterations in round $r$, we conclude that the algorithm is $(\varepsilon, \delta)$-DP.

### B.1.2 Proof of Lemma 9

Recall that we consider the *polyhedral* setup, where the feasible set $\mathcal{W}$ is a polytope with at most $J$ vertices. Since the norm is polyhedral, the dual norm is also polyhedral. Hence, $(\mathbf{E}, \|\cdot\|_*)$ is $(2\log(J))$-regular as discussed earlier in this section.

Fix any $r \in \{0, \ldots, R-1\}$. For any $t \in \{1, \ldots, 2^r - 1\}$, we can write

$$\nabla_r^t - \nabla F_{\mathcal{D}}(w_r^t) = (1 - \eta_{r,t})\left[\nabla_r^{t-1} - \nabla F_{\mathcal{D}}(w_r^{t-1})\right] + (1 - \eta_{r,t})\left[\Delta_r^t - \left(\nabla F_{\mathcal{D}}(w_r^t) - \nabla F_{\mathcal{D}}(w_r^{t-1})\right)\right]$$

$$+ \eta_{r,t}\left[\frac{t+1}{b}\sum_{z \in B_r^t}\nabla f(w_r^t, z) - \nabla F_{\mathcal{D}}(w_r^t)\right].$$

Let $\overline{\Delta}_r^t \triangleq \nabla F_{\mathcal{D}}(w_r^t) - \nabla F_{\mathcal{D}}(w_r^{t-1})$. Recall that $\|\cdot\|_*$ is $(2\log(J))$-regular, and denote $\|\cdot\|_+$ the corresponding $\kappa_+$-smooth norm, where $1 \leq \kappa_+ \leq 2\log(J)$. First we will bound the variance in $\|\cdot\|_+$, and then we will derive the result using the equivalence property (10). Let $\mathcal{Q}_r^t$ be the $\sigma$-algebra generated by the randomness in the data and the algorithm up until iteration $(r, t)$, i.e., the randomness in $\left\{\left(B_k^j, \left(u_k^j(v) : v \in \mathcal{V}\right)\right) : 0 \leq k \leq r, 0 \leq j \leq t\right\}$. Define $\gamma_r^t \triangleq \mathbb{E}\left[\|\nabla_r^t - \nabla F_{\mathcal{D}}(w_r^t)\|_+^2 \mid \mathcal{Q}_r^{t-1}\right]$. By property (9), observe that

$$\gamma_r^t \leq (1 - \eta_{r,t})^2\gamma_r^{t-1} + \kappa_+ \mathbb{E}\left[\left\|(1 - \eta_{r,t})\left(\Delta_r^t - \overline{\Delta}_r^t\right) + \eta_{r,t}\left(\frac{t+1}{b}\sum_{z \in B_r^t}\nabla f(w_r^t, z) - \nabla F_{\mathcal{D}}(w_r^t)\right)\right\|_+^2 \middle| \mathcal{Q}_r^{t-1}\right]$$

$$\leq (1 - \eta_{r,t})^2\gamma_r^{t-1} + 2\kappa_+(1 - \eta_{r,t})^2\mathbb{E}\left[\left\|\Delta_r^t - \overline{\Delta}_r^t\right\|_+^2 \middle| \mathcal{Q}_r^{t-1}\right] + 2\kappa_+\eta_{r,t}^2\mathbb{E}\left[\left\|\frac{t+1}{b}\sum_{z \in B_r^t}\nabla f(w_r^t, z) - \nabla F_{\mathcal{D}}(w_r^t)\right\|_+^2 \middle| \mathcal{Q}_r^{t-1}\right].$$

In the first inequality, we used the fact that $\mathbb{E}_{z \sim \mathcal{D}}[\nabla f(w, z)] = \nabla F_{\mathcal{D}}(w)$, $\mathbb{E}_{z \sim \mathcal{D}}[\Delta_r^t] = \overline{\Delta}_r^t$, and the independence of $\left(\nabla_r^{t-1} - \nabla F_{\mathcal{D}}(w_r^{t-1})\right)$ and $(1 - \eta_{r,t})\left(\Delta_r^t - \overline{\Delta}_r^t\right) + \eta_{r,t}\left(\nabla f(w_r^t, z) - \nabla F_{\mathcal{D}}(w_r^t)\right)$ conditioned on $\mathcal{Q}_r^{t-1}$. The second

inequality follows by triangle inequality and the fact that $(a + b)^2 \leq 2a^2 + 2b^2$ for $a, b \in \mathbb{R}$. Hence, using (10) and $L_1$-smoothness of the loss, we can obtain the following bound inductively:

$$
\begin{aligned}
\mathbb{E}\left[\left\|\Delta_r^t - \overline{\Delta}_r^t\right\|_+^2 \Big| \mathcal{Q}_r^{t-1}\right] &= \mathbb{E}\left[\left\|\frac{t+1}{b} \sum_{z \in B_r^t} \left(\nabla f(w_r^t, z) - \nabla f(w_r^{t-1}, z) - \overline{\Delta}_r^t\right)\right\|_+^2 \Big| \mathcal{Q}_r^{t-1}\right] \\
&\leq \frac{(t+1)^2}{b^2} \mathbb{E}\left[\left\|\sum_{z \in B_r^t \setminus \{z'\}} \left(\nabla f(w_r^t, z) - \nabla f(w_r^{t-1}, z) - \overline{\Delta}_r^t\right)\right\|_+^2 \Big| \mathcal{Q}_r^{t-1}\right] \\
&\quad + \kappa_+ \frac{(t+1)^2}{b^2} \mathbb{E}\left[\left\|\nabla f(w_r^t, z') - \nabla f(w_r^{t-1}, z') - \overline{\Delta}_r^t\right\|_+^2 \Big| \mathcal{Q}_r^{t-1}\right] \\
&\leq \kappa_+ \frac{(t+1)^2}{b^2} \sum_{z \in B_r^t} \mathbb{E}\left[\left\|\nabla f(w_r^t, z) - \nabla f(w_r^{t-1}, z) - \overline{\Delta}_r^t\right\|_+^2 \Big| \mathcal{Q}_r^{t-1}\right] \\
&\leq \kappa \frac{(t+1)^2}{b^2} \sum_{z \in B_r^t} \mathbb{E}\left[\left\|\nabla f(w_r^t, z) - \nabla f(w_r^{t-1}, z) - \overline{\Delta}_r^t\right\|_*^2 \Big| \mathcal{Q}_r^{t-1}\right] \\
&\leq \frac{4 (L_1 D)^2 \log(J) \eta_{r,t}^2 (t+1)}{b},
\end{aligned}
$$

where the inequality before the last one follows from the fact that $\kappa_+ \leq \kappa$, and the last inequality follows from the fact that $\kappa = 2\log(J)$. Similarly, since the loss is $L_0$-Lipschitz, using the same inductive approach, we can bound

$$
\mathbb{E}\left[\left\|\frac{t+1}{b} \sum_{z \in B_r^t} \nabla f(w_r^t, z) - \nabla F_{\mathcal{D}}(w_r^t)\right\|_+^2 \Big| \mathcal{Q}_r^{t-1}\right] \leq \frac{4 L_0^2 \log(J) (t+1)}{b}.
$$

Using the above bounds and the setting of $\eta_{r,t}$, we reach the following recursion

$$
\gamma_r^t \leq \left(1 - \frac{1}{\sqrt{t+1}}\right)^2 \gamma_r^{t-1} + \frac{8\kappa_+ (L_0^2 + L_1^2 D^2) \log(J)}{b}.
$$

Unravelling the recursion, we can further bound $\gamma_r^t$ as:

$$
\begin{aligned}
\gamma_r^t &\leq \gamma_r^0 \left(1 - \frac{1}{\sqrt{t+1}}\right)^{2t} + \frac{8\kappa_+ (L_0^2 + L_1^2 D^2) \log(J)}{b} \sum_{j=0}^{t-1} \left(1 - \frac{1}{\sqrt{t+1}}\right)^{2j} \\
&\leq \gamma_r^0 \left(1 - \frac{1}{\sqrt{t+1}}\right)^{2t} + \frac{8\kappa_+ (L_0^2 + L_1^2 D^2) \log(J) \sqrt{t+1}}{b},
\end{aligned} \tag{12}
$$

where the last inequality follows from the fact that $\sum_{j=0}^{t-1} (1 - \frac{1}{\sqrt{t+1}})^{2j} \leq \frac{1}{1-(1-\frac{1}{\sqrt{t+1}})^2} \leq \sqrt{t+1}$.

Moreover, observe that we can bound $\gamma_r^0$ using the same inductive approach we used earlier:

$$\gamma_r^0 = \mathbb{E}\left[\left\|\frac{1}{b}\sum_{z\in B_r^0}\nabla f(w_r^0, z) - \nabla F_{\mathcal{D}}(w_r^0)\right\|_+^2 \middle| \mathcal{Q}_{r-1}^{2^{r-1}-1}\right]$$

$$\leq \frac{1}{b^2}\left(\mathbb{E}\left[\left\|\sum_{z\in B_r^0\setminus\{z'\}}(\nabla f(w_r^0, z) - \nabla F_{\mathcal{D}}(w_r^0))\right\|_+^2 \middle| \mathcal{Q}_{r-1}^{2^{r-1}-1}\right] + \kappa_+\mathbb{E}\left[\left\|\nabla f(w_r^0, z') - \nabla F_{\mathcal{D}}(w_r^0)\right\|_+^2 \middle| \mathcal{Q}_{r-1}^{2^{r-1}-1}\right]\right)$$

$$\leq \frac{\kappa_+}{b^2}\sum_{z\in B_r^0}\mathbb{E}\left[\left\|\nabla f(w_r^0, z) - \nabla F_{\mathcal{D}}(w_r^0)\right\|_+^2 \middle| \mathcal{Q}_{r-1}^{2^{r-1}-1}\right]$$

$$\leq \frac{4L_0^2\log(J)}{b}.$$

Plugging this in (12), we can finally arrive at

$$\mathbb{E}\left[\left\|\nabla_r^t - \nabla F_{\mathcal{D}}(w_r^t)\right\|_+^2\right] \leq \frac{4L_0^2\log(J)}{b}\left(1 - \frac{1}{\sqrt{t+1}}\right)^{2t} + \frac{8\kappa_+(L_0^2 + L_1^2 D^2)\log(J)\sqrt{t+1}}{b}$$

$$\leq \frac{4L_0^2\log(J)}{b}\left(1 - \frac{1}{\sqrt{t+1}}\right)^{2t} + \frac{16(L_0^2 + L_1^2 D^2)\log^2(J)\sqrt{t+1}}{b},$$

where the last inequality follows from the fact that $\kappa_+ \leq \kappa = 2\log(J)$.

By property (10) of regular norms and using Jensen's inequality together with the subadditivity of the square root, we reach the desired bound:

$$\mathbb{E}\left[\left\|\nabla_r^t - \nabla F_{\mathcal{D}}(w_r^t)\right\|_*\right] \leq \sqrt{\mathbb{E}\left[\left\|\nabla_r^t - \nabla F_{\mathcal{D}}(w_r^t)\right\|_+^2\right]}$$

$$\leq 4L_0\sqrt{\frac{\log(J)}{b}}\left(1 - \frac{1}{\sqrt{t+1}}\right)^t + 4(L_1 D + L_0)\frac{\log(J)}{\sqrt{b}}(t+1)^{1/4}.$$

### B.1.3 Proof of Theorem 8

For any $r \in \{0, \ldots, R-1\}$ and $t \in \{0, \ldots, 2^r - 1\}$, let $\alpha_r^t \triangleq \langle v_r^t, \nabla_r^t\rangle - \min_{v\in\mathcal{V}}\langle v, \nabla_r^t\rangle$; and let $v_{r,t}^* = \arg\min_{v\in\mathcal{W}}\langle\nabla F_{\mathcal{D}}(w_r^t), v - w_r^t\rangle$. By smoothness and convexity of $F_{\mathcal{D}}$, observe

$$F_{\mathcal{D}}(w_r^{t+1}) \leq F_{\mathcal{D}}(w_r^t) + \langle\nabla F_{\mathcal{D}}(w_r^t), w_r^{t+1} - w_r^t\rangle + \frac{L_1}{2}\|w_r^{t+1} - w_r^t\|^2$$

$$\leq F_{\mathcal{D}}(w_r^t) + \eta_{r,t}\langle\nabla F_{\mathcal{D}}(w_r^t) - \nabla_r^t, v_r^t - w_r^t\rangle + \eta_{r,t}\langle\nabla_r^t, v_r^t - w_r^t\rangle + \frac{L_1 D^2\eta_{r,t}^2}{2}$$

$$\leq F_{\mathcal{D}}(w_r^t) + \eta_{r,t}\langle\nabla F_{\mathcal{D}}(w_r^t) - \nabla_r^t, v_r^t - w_r^t\rangle + \eta_{r,t}\langle\nabla_r^t, v_{r,t}^* - w_r^t\rangle + \eta_{r,t}\alpha_r^t + \frac{L_1 D^2\eta_{r,t}^2}{2}$$

$$= F_{\mathcal{D}}(w_r^t) + \eta_{r,t}\langle\nabla F_{\mathcal{D}}(w_r^t) - \nabla_r^t, v_r^t - v_{r,t}^*\rangle - \eta_{r,t}\langle\nabla F_{\mathcal{D}}(w_r^t), v_{r,t}^* - w_r^t\rangle + \eta_{r,t}\alpha_r^t + \frac{L_1 D^2\eta_{r,t}^2}{2}$$

$$\leq F_{\mathcal{D}}(w_r^t) + \eta_{r,t}D\left\|\nabla F_{\mathcal{D}}(w_r^t) - \nabla_r^t\right\|_* - \eta_{r,t}\mathsf{Gap}_{F_{\mathcal{D}}}(w_r^t) + \eta_{r,t}\alpha_r^t + \frac{L_1 D^2\eta_{r,t}^2}{2}.$$

Hence, we have

$$\mathbb{E}[\mathsf{Gap}_{F_{\mathcal{D}}}(w_r^t)] \leq \frac{\mathbb{E}[F_{\mathcal{D}}(w_r^t) - F_{\mathcal{D}}(w_r^{t+1})]}{\eta_{r,t}} + \frac{L_1 D^2\eta_{r,t}}{2} + D\,\mathbb{E}\left[\left\|\nabla_r^t - \nabla F_{\mathcal{D}}(w_r^t)\right\|_*\right] + \mathbb{E}[\alpha_r^t].$$

Note that by a standard argument $\mathbb{E}\left[\alpha_r^t\right] \leq 2s_r\log(J) = \frac{4D(L_0+L_1D)2^r\log(J)\sqrt{\log(1/\delta)}}{b\varepsilon}$. Thus, given the bound on $\mathbb{E}\left[\|\nabla_r^t - \nabla F_{\mathcal{D}}(w_r^t)\|_*\right]$ from Lemma 9, we have

$$\mathbb{E}[\mathsf{Gap}_{F_{\mathcal{D}}}(w_r^t)] \leq \sqrt{t+1}\left(\mathbb{E}[F_{\mathcal{D}}(w_r^t) - F_{\mathcal{D}}(w_r^{t+1})]\right) + \frac{L_1D^2}{2\sqrt{t+1}} + 4L_0D\sqrt{\frac{\log(J)}{b}}\left(1 - \frac{1}{\sqrt{t+1}}\right)^t$$
$$+ 4D\left(L_1D + L_0\right)\frac{\log(J)}{\sqrt{b}}(t+1)^{1/4} + 4D(L_0 + L_1D)\frac{\log(J)\sqrt{\log(1/\delta)}}{b\varepsilon}2^r.$$

For any given $r \in \{0, \ldots, R-1\}$, we now sum both sides of the above inequality over $t \in \{0, \ldots, 2^r - 1\}$.

Let $\Gamma_r \triangleq \sum_{t=0}^{2^r-1}\sqrt{t+1}\left(\mathbb{E}[F_{\mathcal{D}}(w_r^t) - F_{\mathcal{D}}(w_r^{t+1})]\right)$. Observe that

$$\sum_{t=0}^{2^r-1}\mathbb{E}[\mathsf{Gap}_{F_{\mathcal{D}}}(w_r^t)] \leq \Gamma_r + \frac{L_1D^2}{2}\sum_{t=1}^{2^r}\frac{1}{\sqrt{t}} + 4L_0D\sqrt{\frac{\log(J)}{b}}\sum_{t=0}^{2^r-1}\left(1 - \frac{1}{\sqrt{t+1}}\right)^t$$
$$+ 4D(L_0 + DL_1)\frac{\log(J)}{\sqrt{b}}\sum_{t=1}^{2^r}t^{1/4} + 4D(L_0 + L_1D)\frac{\log(J)\sqrt{\log(1/\delta)}}{b\varepsilon}2^{2r}$$
$$\leq \Gamma_r + L_1D^2\,2^{r/2} + 4L_0D\sqrt{\frac{\log(J)}{b}}\sum_{t=0}^{2^r-1}(1 - 2^{-r/2})^t$$
$$+ 8D(L_0 + DL_1)\frac{\log(J)}{\sqrt{b}}2^{5r/4} + 4D(L_0 + L_1D)\frac{\log(J)\sqrt{\log(1/\delta)}}{b\varepsilon}2^{2r}$$
$$\leq \Gamma_r + L_1D^2 2^{r/2} + 4L_0D\sqrt{\frac{\log(J)}{b}}2^{r/2} + 8D(L_0 + L_1D)\frac{\log(J)}{\sqrt{b}}2^{5r/4}$$
$$+ 4D(L_0 + L_1D)\frac{\log(J)\sqrt{\log(1/\delta)}}{b\varepsilon}2^{2r}.$$

Next, we bound $\Gamma_r$. Before we do so, note that for all $z \in \mathcal{Z}$, $f(\cdot, z)$ is $L_0$-Lipschitz and the $\|\cdot\|$-diameter of $\mathcal{W}$ is bounded by $D$, hence, w.l.o.g., we will assume that the range of $f(\cdot, z)$ lies in $[-L_0D, L_0D]$. This implies that the range of $F_{\mathcal{D}}$ lies in $[-L_0D, L_0D]$. Now, observe that

$$\Gamma_r = \sum_{t=0}^{2^r-1}\sqrt{t+1}\ \left(\mathbb{E}[F_{\mathcal{D}}(w_r^t) - F_{\mathcal{D}}(w_r^{t+1})]\right)$$
$$= \sum_{t=0}^{2^r-1}\left(\sqrt{t+1}\ \mathbb{E}\left[F_{\mathcal{D}}(w_r^t)\right] - \sqrt{t+2}\ \mathbb{E}\left[F_{\mathcal{D}}(w_r^{t+1})\right]\right) + \sum_{t=0}^{2^r-1}\left(\sqrt{t+2} - \sqrt{t+1}\right)\mathbb{E}\left[F_{\mathcal{D}}(w_r^{t+1})\right]$$
$$\leq \sum_{t=0}^{2^r-1}\left(\sqrt{t+1}\ \mathbb{E}\left[F_{\mathcal{D}}(w_r^t)\right] - \sqrt{t+2}\ \mathbb{E}\left[F_{\mathcal{D}}(w_r^{t+1})\right]\right) + L_0D\sum_{t=0}^{2^r-1}\left(\sqrt{t+2} - \sqrt{t+1}\right)$$

Note that both sums on the right-hand side are telescopic. Hence, we get

$$\Gamma_r \leq \mathbb{E}\left[F_{\mathcal{D}}(w_r^0) - \sqrt{2^r+1}F_{\mathcal{D}}(w_r^{2^r})\right] + L_0D\,2^{r/2}$$
$$= \mathbb{E}\left[F_{\mathcal{D}}(w_r^0) - F_{\mathcal{D}}(w_r^{2^r})\right] - \left(\sqrt{2^r+1} - 1\right)\mathbb{E}\left[F_{\mathcal{D}}(w_r^{2^r})\right] + L_0D\,2^{r/2}$$
$$\leq 3L_0D\,2^{r/2}.$$

Thus, we arrive at

$$\sum_{t=0}^{2^r-1} \mathbb{E}[\mathsf{Gap}_{F_{\mathcal{D}}}(w_r^t)] \le 3D(L_0+L_1 D)2^{r/2} + 4L_0 D\sqrt{\frac{\log(J)}{b}}2^{r/2} + 8D(L_0+L_1 D)\frac{\log(J)}{\sqrt{b}}2^{5r/4}$$

$$+ 4D(L_0+L_1 D)\frac{\log(J)\sqrt{\log(1/\delta)}}{b\varepsilon}2^{2r}.$$

Now, summing over all rounds $r \in \{0,\dots,R-1\}$, we have

$$\sum_{r=0}^{R-1}\sum_{t=0}^{2^r-1} \mathbb{E}[\mathsf{Gap}_{F_{\mathcal{D}}}(w_r^t)] \le 9D(L_0+L_1 D)2^{R/2} + 12L_0 D\sqrt{\frac{\log(J)}{b}}2^{R/2} + 6D(L_0+L_1 D)\frac{\log(J)}{\sqrt{b}}2^{5R/4}$$

$$+ 2D(L_0+L_1 D)\frac{\log(J)\sqrt{\log(1/\delta)}}{b\varepsilon}2^{2R}.$$

Recall that the output $\widehat{w}$ is uniformly chosen from the set of all $2^R$ iterates. By taking expectation with respect to that random choice and using the above, we get

$$\mathbb{E}[\mathsf{Gap}_{F_{\mathcal{D}}}(\widehat{w})] = \frac{1}{2^R}\sum_{r=0}^{R-1}\sum_{t=0}^{2^r-1} \mathbb{E}[\mathsf{Gap}_{F_{\mathcal{D}}}(w_r^t)]$$

$$\le 9D(L_0+L_1 D)2^{-R/2} + 12L_0 D\sqrt{\frac{\log(J)}{b}}2^{-R/2} + 6D(L_0+L_1 D)\frac{\log(J)}{\sqrt{b}}2^{R/4}$$

$$+ 2D(L_0+L_1 D)\frac{\log(J)\sqrt{\log(1/\delta)}}{b\varepsilon}2^{R}.$$

Recall that $R = \frac{2}{3}\log\left(\frac{n\varepsilon}{\log^2(J)\log^2(n)\sqrt{\log(1/\delta)}}\right)$ and $b = \frac{n}{\log^2(n)}$. Hence, we have

$$\mathbb{E}[\mathsf{Gap}_{F_{\mathcal{D}}}(\widehat{w})] \le 9D(L_0+L_1 D)\left(\frac{\log^2(J)\sqrt{\log(1/\delta)}\log^2(n)}{n\varepsilon}\right)^{1/3} + 12L_0 D\sqrt{\frac{\log(J)\log^2(n)}{n}}\left(\frac{\log^2(J)\sqrt{\log(1/\delta)}\log^2(n)}{n\varepsilon}\right)^{1/3}$$

$$+ 6D(L_0+L_1 D)\frac{\varepsilon^{1/6}}{\log^{1/3}(n)\log^{1/12}(1/\delta)}\left(\frac{\log^2(J)}{n}\right)^{1/3} + 2D(L_0+L_1 D)\left(\frac{\log^2(J)\sqrt{\log(1/\delta)}\log^2(n)}{n\varepsilon}\right)^{1/3}$$

$$= O\left(D(L_0+L_1 D)\left(\frac{\log^2(J)\log^2(n)\sqrt{\log(1/\delta)}}{n\varepsilon}\right)^{1/3}\right),$$

which is the claimed bound.

## B.2 Missing Details of Section 4.2

### B.2.1 Noisy Stochastic Frank-Wolfe

A formal description of the noisy stochastic Frank-Wolfe algorithm for non-convex smooth losses in the $\ell_p$ setting is given in Algorithm 6 below.

**Algorithm 6** $\mathcal{A}_{\mathsf{nSFW}}$: Private Noisy Stochastic Frank-Wolfe Algorithm for $\ell_p$ DP-SO, $1 < p \le 2$

---

**Require:** Private dataset $S = (z_1, \ldots z_n) \in \mathcal{Z}^n$, privacy parameters $(\varepsilon, \delta)$, a number $p \in (1, 2]$ feasible set $\mathcal{W} \subset \mathbb{R}^d$ with $\|\cdot\|_p$-diameter $D$, number of rounds $R$, batch size $b$, step sizes $(\eta_{r,t} : r = 0, \ldots, R-1, \ t = 0, \ldots, 2^r - 1)$

1: Choose an arbitrary initial point $w_0^0 \in \mathcal{W}$
2: **for** $r = 0$ to $R - 1$ **do**
3:     Let $\sigma_{r,0}^2 = \frac{16L_0^2 d^{2/p-1} \log(1/\delta)}{b^2 \varepsilon^2}$
4:     Draw a batch $B_r^0$ of $b$ samples without replacement from $S$
5:     Compute $\widetilde{\nabla}_r^0 = \frac{1}{b} \sum_{z \in B_r^0} \nabla f(w_r^0, z) + N_r^0, \ \ N_r^0 \sim \mathcal{N}\left(0, \sigma_{r,0}^2 \mathbb{I}_d\right)$
6:     $v_r^0 = \arg\min_{v \in \mathcal{W}} \langle v, \widetilde{\nabla}_r^0 \rangle$
7:     $w_r^1 \leftarrow (1 - \eta_{r,0}) w_r^0 + \eta_{r,0} v_r^0$
8:     **for** $t = 1$ to $2^r - 1$ **do**
9:         Let $\sigma_{r,t}^2 = \frac{16L_0^2 (t+1)^2 d^{2/p-1} \log(1/\delta)}{b^2 \varepsilon^2}, \ \widehat{\sigma}_{r,t}^2 = \frac{16L_1^2 D^2 \eta_{r,t}^2 (t+1)^2 d^{2/p-1} \log(1/\delta)}{b^2 \varepsilon^2}$
10:         Draw a batch $B_r^t$ of $b/(t+1)$ samples without replacement from $S$
11:         Let $\Delta_r^t = \frac{t+1}{b} \sum_{z \in B_r^t} \left( \nabla f(w_r^t, z) - \nabla f(w_r^{t-1}, z) \right)$, and let $g_r^t = \frac{t+1}{b} \sum_{z \in B_r^t} \nabla f(w_r^t, z)$
12:         Compute $\widetilde{\Delta}_r^t = \Delta_r^t + \widehat{N}_r^t, \ \ \widehat{N}_r^t \sim \mathcal{N}\left(0, \widehat{\sigma}_{r,t}^2 \mathbb{I}_d\right)$
13:         Compute $\widetilde{g}_r^t = g_r^t + N_r^t, \ \ N_r^t \sim \mathcal{N}\left(0, \sigma_{r,t}^2 \mathbb{I}_d\right)$
14:         $\widetilde{\nabla}_r^t = (1 - \eta_{r,t}) \left( \widetilde{\nabla}_r^{t-1} + \widetilde{\Delta}_r^t \right) + \eta_{r,t} \widetilde{g}_r^t$
15:         Compute $v_r^t = \arg\min_{v \in \mathcal{W}} \langle v, \widetilde{\nabla}_r^t \rangle$
16:         $w_r^{t+1} \leftarrow (1 - \eta_{r,t}) w_r^t + \eta_{r,t} v_r^t$
17:     $w_{r+1}^0 = w_r^{2^r}$
18: Output $\widehat{w}$ uniformly chosen from the set of all iterates $(w_r^t : r = 0, \ldots, R-1, t = 0, \ldots, 2^r - 1)$

---

### B.2.2    Proof of Theorem 10

Note that it suffices to show that for any given $(r, t)$, $r \in \{0, \ldots, R-1\}$, $t \in [2^r - 1]$, computing $\widetilde{\nabla}_r^0$ (Step 5 in Algorithm 6) satisfies $(\varepsilon, \delta)$-DP, and computing $\widetilde{\Delta}_r^t, \widetilde{g}_r^t$ (Steps 12 and 13) satisfies $(\varepsilon, \delta)$-DP. Assuming we can show that this is the case, then note that at any given iteration $(r, t)$, the gradient estimate $\widetilde{\nabla}_r^{t-1}$ from the previous iteration is already computed privately. Since differential privacy is closed under post-processing, then the current iteration is also $(\varepsilon, \delta)$-DP. Since the batches used in different iterations are disjoint, then by parallel composition, the algorithm is $(\varepsilon, \delta)$-DP. Thus, it remains to show that for any given $(r, t)$, the steps mentioned above are computed in $(\varepsilon, \delta)$-DP manner. Let $S, S'$ be neighboring datasets (i.e., differing in exactly one point). Let $\widetilde{\nabla}_r^0, \widetilde{\Delta}_r^t, \widetilde{g}_r^t$ be the quantities above when the input dataset is $S$; and let $\widetilde{\nabla}_r'^0, \widetilde{\Delta}_r'^t, \widetilde{g}_r'^t$ be the corresponding quantities when the input dataset is $S'$. Note that the $\ell_2$-sensitivity of $\widetilde{\nabla}_r^0$ can be bounded as $\left\| \widetilde{\nabla}_r^0 - \widetilde{\nabla}_r'^0 \right\|_2 \le d^{\frac{1}{p} - \frac{1}{2}} \left\| \widetilde{\nabla}_r^0 - \widetilde{\nabla}_r'^0 \right\|_* \le \frac{L_0 d^{\frac{1}{p} - \frac{1}{2}}}{b}$, where the dual norm here is $\|\cdot\|_* = \|\cdot\|_q$ where $q = \frac{p}{p-1}$. Similarly, we can bound the $\ell_2$-sensitivity of $\widetilde{g}_r^t$ as $\left\| \widetilde{g}_r^t - \widetilde{g}_r'^t \right\|_2 \le \frac{L_0 d^{\frac{1}{p} - \frac{1}{2}} (t+1)}{b}$. Also, by the $L_1$-smoothness of the loss, we have $\left\| \widetilde{\Delta}_r^t - \widetilde{\Delta}_r'^t \right\|_2 \le d^{\frac{1}{p} - \frac{1}{2}} \left\| \widetilde{\Delta}_r^t - \widetilde{\Delta}_r'^t \right\|_* \le \frac{L_1 D \eta_{r,t} d^{\frac{1}{p} - \frac{1}{2}} (t+1)}{b}$. Given these bounds and the settings of the noise parameters in the algorithm, the argument follows directly by the privacy guarantee of the Gaussian mechanism.

### B.2.3 Proof of Lemma 12

Note that for the $\ell_p$ space, where $p \in (1, 2]$, the dual is the $\ell_q$ space where $q = \frac{p}{p-1} \geq 2$. To keep the notation consistent with the rest of the paper, in the sequel, we will be using $\|\cdot\|_*$ to denote the dual norm $\|\cdot\|_q$ unless specific reference to $q$ is needed. As discussed earlier in this section, the dual space $\ell_q$ is $\kappa$-regular with $\kappa = \min\left(q - 1, 2\log(d)\right) = \min\left(\frac{1}{p-1}, 2\log(d)\right)$.

Fix any $r \in \{0, \ldots, R-1\}$ and $t \in \{1, \ldots, 2^r - 1\}$. As we did in the proof of Lemma 9, we write

$$\widetilde{\nabla}_r^t - \nabla F_{\mathcal{D}}(w_r^t) = (1 - \eta_{r,t}) \left[\widetilde{\nabla}_r^{t-1} - \nabla F_{\mathcal{D}}(w_r^{t-1})\right] + (1 - \eta_{r,t}) \left[\widetilde{\Delta}_r^t - \overline{\Delta}_r^t\right]$$
$$+ \eta_{r,t} \left[\widetilde{g}_r^t - \nabla F_{\mathcal{D}}(w_r^t)\right].$$

where $\overline{\Delta}_r^t \triangleq \nabla F_{\mathcal{D}}(w_r^t) - \nabla F_{\mathcal{D}}(w_r^{t-1})$.

Let $\|\cdot\|_+$ denote the $\kappa_+$-smooth norm associated with $\|\cdot\|_*$ (as defined by the regularity property, in the beginning of this section). Note that by $\kappa$-regularity of $\|\cdot\|_*$, such norm exists for some $1 \leq \kappa_+ \leq \kappa$. Let $\mathcal{Q}_r^t$ be the $\sigma$-algebra induced by all the randomness up until the iteration indexed by $(r, t)$. Define $\gamma_r^t \triangleq \mathbb{E}\left[\left\|\widetilde{\nabla}_r^t - \nabla F_{\mathcal{D}}(w_r^t)\right\|_+^2 \mid \mathcal{Q}_r^{t-1}\right]$. Note by property (9) of $\kappa$-regular norms, we have

$$\gamma_r^t \leq (1 - \eta_{r,t})^2 \gamma_r^{t-1} + \kappa_+ \mathbb{E}\left[\left\|(1 - \eta_{r,t})\left(\widetilde{\Delta}_r^t - \overline{\Delta}_r^t\right) + \eta_{r,t}\left(\widetilde{g}_r^t - \nabla F_{\mathcal{D}}(w_r^t)\right)\right\|_+^2 \mid \mathcal{Q}_r^{t-1}\right]$$

$$\leq (1 - \eta_{r,t})^2 \gamma_r^{t-1} + \kappa_+ \mathbb{E}\left[\left\|(1 - \eta_{r,t})\left(\Delta_r^t - \overline{\Delta}_r^t + \widehat{N}_r^t\right) + \eta_{r,t}\left(g_r^t - \nabla F_{\mathcal{D}}(w_r^t) + N_r^t\right)\right\|_+^2 \mid \mathcal{Q}_r^{t-1}\right]$$

$$\leq (1 - \eta_{r,t})^2 \gamma_r^{t-1} + 2\kappa_+(1 - \eta_{r,t})^2 \mathbb{E}\left[\left\|\Delta_r^t - \overline{\Delta}_r^t + \widehat{N}_r^t\right\|_+^2 \mid \mathcal{Q}_r^{t-1}\right] + 2\kappa_+ \eta_{r,t}^2 \mathbb{E}\left[\left\|g_r^t - \nabla F_{\mathcal{D}}(w_r^t) + N_r^t\right\|_+^2 \mid \mathcal{Q}_r^{t-1}\right]$$

$$\leq (1 - \eta_{r,t})^2 \gamma_r^{t-1} + 4\kappa_+(1 - \eta_{r,t})^2 \mathbb{E}\left[\left\|\Delta_r^t - \overline{\Delta}_r^t\right\|_+^2 \mid \mathcal{Q}_r^{t-1}\right] + 4\kappa_+(1 - \eta_{r,t})^2 \mathbb{E}\left[\left\|\widehat{N}_r^t\right\|_+^2 \mid \mathcal{Q}_r^{t-1}\right]$$

$$+ 4\kappa_+ \eta_{r,t}^2 \mathbb{E}\left[\left\|g_r^t - \nabla F_{\mathcal{D}}(w_r^t)\right\|_+^2 \mid \mathcal{Q}_r^{t-1}\right] + 4\kappa_+ \eta_{r,t}^2 \mathbb{E}\left[\left\|N_r^t\right\|_+^2 \mid \mathcal{Q}_r^{t-1}\right]. \tag{13}$$

where the last two inequalities follow from the triangle inequality.

Now, using the same inductive approach we used in the proof of Lemma 9, we can bound

$$\mathbb{E}\left[\left\|\Delta_r^t - \overline{\Delta}_r^t\right\|_+^2 \mid \mathcal{Q}_r^{t-1}\right] \leq \kappa \frac{(t+1)^2}{b^2} \sum_{z \in B_r^t} \mathbb{E}\left[\left\|\nabla f(w_r^t, z) - \nabla f(w_r^{t-1}, z) - \overline{\Delta}_r^t\right\|_*^2 \mid \mathcal{Q}_r^{t-1}\right] \leq \frac{2\kappa L_1^2 D^2 \eta_{r,t}^2 (t+1)}{b},$$

$$\mathbb{E}\left[\left\|g_r^t - \nabla F_{\mathcal{D}}(w_r^t)\right\|_+^2 \mid \mathcal{Q}_r^{t-1}\right] \leq \kappa \frac{(t+1)^2}{b^2} \sum_{z \in B_r^t} \mathbb{E}\left[\left\|\nabla f(w_r^t, z) - \nabla F_{\mathcal{D}}(w_r^t)\right\|_*^2 \mid \mathcal{Q}_r^{t-1}\right] \leq \frac{2\kappa L_0^2 (t+1)}{b}$$

Moreover, observe that by property (10) of $\kappa$-regular norms, we have

$$\mathbb{E}\left[\left\|\widehat{N}_r^t\right\|_+^2 \mid \mathcal{Q}_r^{t-1}\right] \leq \frac{\kappa}{\kappa_+} \mathbb{E}\left[\left\|\widehat{N}_r^t\right\|_*^2 \mid \mathcal{Q}_r^{t-1}\right] = \frac{\kappa}{\kappa_+} \mathbb{E}\left[\left\|\widehat{N}_r^t\right\|_q^2 \mid \mathcal{Q}_r^{t-1}\right]$$

Note that when $p = q = 2$ (i.e., the Euclidean setting), then the above is bounded by $d\widehat{\sigma}_{r,t}^2$ (in such case, note that $\kappa = \kappa_+ = 1$). Otherwise (when $1 < p < 2$), we have

$$\mathbb{E}\left[\left\|\widehat{N}_r^t\right\|_+^2 \middle| \mathcal{Q}_r^{t-1}\right] \leq \frac{\kappa}{\kappa_+}\mathbb{E}\left[\left\|\widehat{N}_r^t\right\|_*^2 \middle| \mathcal{Q}_r^{t-1}\right] = \frac{\kappa}{\kappa_+}\mathbb{E}\left[\left\|\widehat{N}_r^t\right\|_q^2 \middle| \mathcal{Q}_r^{t-1}\right]$$

$$\leq \frac{\kappa}{\kappa_+}d^{\frac{2}{q}}\mathbb{E}\left[\left\|\widehat{N}_r^t\right\|_\infty^2 \middle| \mathcal{Q}_r^{t-1}\right]$$

$$\leq 2\frac{\kappa}{\kappa_+}d^{\frac{2}{q}}\log(d)\,\widehat{\sigma}_{r,t}^2$$

$$= 32\frac{\kappa}{\kappa_+}\frac{L_1^2 D^2 \eta_{r,t}^2 (t+1)^2\, d\log(d)\log(1/\delta)}{b^2\varepsilon^2}$$

Hence, putting the above together, for any $p \in (1, 2]$, we have

$$\mathbb{E}\left[\left\|\widehat{N}_r^t\right\|_+^2 \middle| \mathcal{Q}_r^{t-1}\right] \leq 32\frac{\kappa\widetilde{\kappa}}{\kappa_+}\frac{L_1^2 D^2 \eta_{r,t}^2 (t+1)^2\, d\log(1/\delta)}{b^2\varepsilon^2},$$

where $\widetilde{\kappa} = 1 + \log(d) \cdot \mathbf{1}(p < 2)$.

Similarly, we can show

$$\mathbb{E}\left[\left\|N_r^t\right\|_+^2 \middle| \mathcal{Q}_r^{t-1}\right] \leq 2\frac{\kappa\widetilde{\kappa}}{\kappa_+}d^{\frac{2}{q}}\sigma_{r,t}^2 = 32\frac{\kappa\widetilde{\kappa}}{\kappa_+}\frac{L_0^2 (t+1)^2\, d\log(1/\delta)}{b^2\varepsilon^2}.$$

Plugging these bounds in inequality (13) and using the setting of $\eta_{r,t}$ in the lemma statement, we arrive at the following recursion:

$$\gamma_r^t \leq \left(1 - \frac{1}{\sqrt{t+1}}\right)^2 \gamma_r^{t-1} + 8\frac{\kappa\kappa_+(L_0^2 + L_1^2 D^2)}{b} + 128\frac{\kappa\widetilde{\kappa}(L_0^2 + L_1^2 D^2)(t+1)d\log(1/\delta)}{b^2\varepsilon^2}$$

$$\leq \left(1 - \frac{1}{\sqrt{t+1}}\right)^2 \gamma_r^{t-1} + 8\frac{\kappa^2(L_0^2 + L_1^2 D^2)}{b} + 128\frac{\kappa\widetilde{\kappa}(L_0^2 + L_1^2 D^2)(t+1)d\log(1/\delta)}{b^2\varepsilon^2},$$

where the last inequality follows from the fact that $\kappa_+ \leq \kappa$. Unraveling this recursion similar to what we did in the proof of Lemma 9, we arrive at

$$\gamma_r^t \leq \left(1 - \frac{1}{\sqrt{t+1}}\right)^{2t} \gamma_r^0 + \left(8\frac{\kappa^2(L_0^2 + L_1^2 D^2)}{b} + 128\frac{\kappa\widetilde{\kappa}(L_0^2 + L_1^2 D^2)(t+1)d\log(1/\delta)}{b^2\varepsilon^2}\right)\sqrt{t+1}. \qquad (14)$$

Now, we can bound $\gamma_r^0$ via the same approach used before:

$$\gamma_r^0 = \mathbb{E}\left[\left\|\frac{1}{b}\sum_{z\in B_r^0}\nabla f(w_r^0,z) - \nabla F_{\mathcal{D}}(w_r^0) + N_r^0\right\|_+^2 \middle| \mathcal{Q}_{r-1}^{2^{r-1}-1}\right]$$

$$\leq 2\,\mathbb{E}\left[\left\|\frac{1}{b}\sum_{z\in B_r^0}\nabla f(w_r^0,z) - \nabla F_{\mathcal{D}}(w_r^0)\right\|_+^2 \middle| \mathcal{Q}_{r-1}^{2^{r-1}-1}\right] + 2\,\mathbb{E}\left[\left\|N_r^0\right\|_+^2 \middle| \mathcal{Q}_{r-1}^{2^{r-1}-1}\right]$$

$$\leq 2\frac{\kappa}{b^2}\sum_{z\in B_r^0}\mathbb{E}\left[\left\|\nabla f(w_r^0,z) - \nabla F_{\mathcal{D}}(w_r^0)\right\|_+^2 \middle| \mathcal{Q}_{r-1}^{2^{r-1}-1}\right] + 64\frac{\kappa\widetilde\kappa}{\kappa_+}\frac{L_0^2 d\log(1/\delta)}{b^2\varepsilon^2}$$

$$\leq 4\frac{\kappa L_0^2}{b} + 64\frac{\kappa\widetilde\kappa}{\kappa_+}\frac{L_0^2 d\log(1/\delta)}{b^2\varepsilon^2}$$

$$\leq 4\frac{\kappa L_0^2}{b} + 64\frac{\kappa\widetilde\kappa L_0^2 d\log(1/\delta)}{b^2\varepsilon^2},$$

where the last inequality follows from the fact that $\kappa_+ \geq 1$. Plugging this in (14), we finally have

$$\mathbb{E}\left[\left\|\widetilde\nabla_r^t - \nabla F_{\mathcal{D}}(w_r^t)\right\|_+^2\right] \leq 64L_0^2\left(\frac{\kappa}{b} + \frac{\kappa\widetilde\kappa d\log(1/\delta)}{b^2\varepsilon^2}\right)\left(1 - \frac{1}{\sqrt{t+1}}\right)^{2t}$$
$$+ 128(L_0^2 + L_1^2 D^2)\left(\frac{\kappa^2}{b}\sqrt{t+1} + \frac{\kappa\widetilde\kappa d\log(1/\delta)}{b^2\varepsilon^2}(t+1)^{3/2}\right).$$

Hence, by property (10) of $\kappa$-regular norms and using Jensen's inequality together with the subadditivity of the square root, we conclude

$$\mathbb{E}\left[\left\|\widetilde\nabla_r^t - \nabla F_{\mathcal{D}}(w_r^t)\right\|_*\right] \leq \sqrt{\mathbb{E}\left[\left\|\widetilde\nabla_r^t - \nabla F_{\mathcal{D}}(w_r^t)\right\|_+^2\right]}$$

$$\leq 8L_0\left(\sqrt{\frac{\kappa}{b}} + \frac{\sqrt{\kappa\widetilde\kappa d\log(1/\delta)}}{b\varepsilon}\right)\left(1 - \frac{1}{\sqrt{t+1}}\right)^t + 16(L_0 + L_1 D)\left(\frac{\kappa}{\sqrt{b}}(t+1)^{1/4} + \frac{\sqrt{\kappa\widetilde\kappa d\log(1/\delta)}}{b\varepsilon}(t+1)^{3/4}\right).$$

### B.2.4  Proof of Theorem 11

For any iteration $(r,t)$, using the same derivation approach as in the proof of Theorem 8, we arrive at the following bound:

$$F_{\mathcal{D}}(w_r^t) \leq F_{\mathcal{D}}(w_r^t) + \eta_{r,t}D\left\|\nabla F_{\mathcal{D}}(w_r^t) - \nabla_r^t\right\|_* - \eta_{r,t}\mathsf{Gap}_{F_{\mathcal{D}}}(w_r^t) + \frac{L_1 D^2\eta_{r,t}^2}{2}$$

Thus, using the bound of Lemma 12, the expected stationarity gap of any given iterate $w_r^t$ can be bounded as:

$$\mathbb{E}[\mathsf{Gap}_{F_{\mathcal{D}}}(w_r^t)] \leq \frac{\mathbb{E}[F_{\mathcal{D}}(w_r^t) - F_{\mathcal{D}}(w_r^{t+1})]}{\eta_{r,t}} + D\,\mathbb{E}\left[\left\|\nabla_r^t - \nabla F_{\mathcal{D}}(w_r^t)\right\|_*\right] + \frac{L_1 D^2\eta_{r,t}}{2}$$

$$\leq \sqrt{t+1}\left(\mathbb{E}[F_{\mathcal{D}}(w_r^t) - F_{\mathcal{D}}(w_r^{t+1})]\right) + \frac{L_1 D^2}{2\sqrt{t+1}} + 8DL_0\left(\sqrt{\frac{\kappa}{b}} + \frac{\sqrt{d\kappa\widetilde\kappa\log(1/\delta)}}{b\varepsilon}\right)\left(1 - 2^{-r/2}\right)^t$$

$$+ 16D\left(L_1 D + L_0\right)\left(\frac{\kappa}{\sqrt{b}}(t+1)^{1/4} + \frac{\sqrt{d\kappa\widetilde\kappa\log(1/\delta)}}{b\varepsilon}(t+1)^{3/4}\right).$$

For any given $r \in \{0, \ldots, R-1\}$, we now sum both sides of the above inequality over $t \in \{0, \ldots, 2^r - 1\}$ as we did in the proof of Theorem 8. Let $\Gamma_r \triangleq \sum_{t=0}^{2^r-1} \sqrt{t+1} \left( \mathbb{E}[F_{\mathcal{D}}(w_r^t) - F_{\mathcal{D}}(w_r^{t+1})] \right)$. Observe that

$$
\begin{aligned}
\sum_{t=0}^{2^r-1} \mathbb{E}[\mathsf{Gap}_{F_{\mathcal{D}}}(w_r^t)] &\leq \Gamma_r + \frac{L_1 D^2}{2} \sum_{t=1}^{2^r} \frac{1}{\sqrt{t}} + 8DL_0 \left( \sqrt{\frac{\kappa}{b}} + \frac{\sqrt{d\kappa\widetilde{\kappa}\log(1/\delta)}}{b\varepsilon} \right) \sum_{t=0}^{2^r-1} \left( 1 - 2^{-r/2} \right)^t \\
&\quad + 16D(L_1 D + L_0) \left( \frac{\kappa}{\sqrt{b}} \sum_{t=1}^{2^r} t^{1/4} + \frac{\sqrt{d\kappa\widetilde{\kappa}\log(1/\delta)}}{b\varepsilon} \sum_{t=1}^{2^r} t^{3/4} \right) \\
&\leq \Gamma_r + L_1 D^2 \, 2^{r/2} + 8DL_0 \left( \sqrt{\frac{\kappa}{b}} + \frac{\sqrt{d\kappa\widetilde{\kappa}\log(1/\delta)}}{b\varepsilon} \right) 2^{r/2} \\
&\quad + 32D(L_1 D + L_0) \left( \frac{\kappa}{\sqrt{b}} 2^{5r/4} + \frac{\sqrt{d\kappa\widetilde{\kappa}\log(1/\delta)}}{b\varepsilon} 2^{7r/4} \right).
\end{aligned}
$$

Next, using exactly the same technique we used in the proof of Theorem 8, we can bound $\Gamma_r \leq 3L_0 D \, 2^{r/2}$. Thus, we arrive at

$$
\begin{aligned}
\sum_{t=0}^{2^r-1} \mathbb{E}[\mathsf{Gap}_{F_{\mathcal{D}}}(w_r^t)] &\leq 3D(L_0 + L_1 D) \, 2^{r/2} + 8DL_0 \left( \sqrt{\frac{\kappa}{b}} + \frac{\sqrt{d\kappa\widetilde{\kappa}\log(1/\delta)}}{b\varepsilon} \right) 2^{r/2} \\
&\quad + 32D(L_1 D + L_0) \left( \frac{\kappa}{\sqrt{b}} 2^{5r/4} + \frac{\sqrt{d\kappa\widetilde{\kappa}\log(1/\delta)}}{b\varepsilon} 2^{7r/4} \right)
\end{aligned}
$$

Now, summing over $r \in \{0, \ldots, R-1\}$, we have

$$
\begin{aligned}
\sum_{r=0}^{R-1} \sum_{t=0}^{2^r-1} \mathbb{E}[\mathsf{Gap}_{F_{\mathcal{D}}}(w_r^t)] &\leq 9D(L_0 + L_1 D) \, 2^{R/2} + 24DL_0 \left( \sqrt{\frac{\kappa}{b}} + \frac{\sqrt{d\kappa\widetilde{\kappa}\log(1/\delta)}}{b\varepsilon} \right) 2^{R/2} \\
&\quad + 48D(L_1 D + L_0) \frac{\kappa}{\sqrt{b}} 2^{5R/4} + 24D(L_1 D + L_0) \frac{\sqrt{d\kappa\widetilde{\kappa}\log(1/\delta)}}{b\varepsilon} 2^{7R/4}.
\end{aligned}
$$

Since the output $\widehat{w}$ is uniformly chosen from the set of all $2^R$ iterates, then averaging over all the iterates gives the following (after some algebra similar to what we did in the proof of Theorem 8)

$$
\begin{aligned}
\mathbb{E}[\mathsf{Gap}_{F_{\mathcal{D}}}(\widehat{w})] = \frac{1}{2^R} \sum_{r=0}^{R-1} \sum_{t=0}^{2^r-1} \mathbb{E}[\mathsf{Gap}_{F_{\mathcal{D}}}(w_r^t)] &\leq 9D(L_0 + L_1 D) 2^{-R/2} + 24DL_0 \left( \sqrt{\frac{\kappa}{b}} + \frac{\sqrt{d\kappa\widetilde{\kappa}\log(1/\delta)}}{b\varepsilon} \right) 2^{-R/2} \\
&\quad + 48D(L_0 + L_1 D) \frac{\kappa}{\sqrt{b}} 2^{R/4} + 24D(L_0 + L_1 D) \frac{\sqrt{d\kappa\widetilde{\kappa}\log(1/\delta)}}{b\varepsilon} 2^{3R/4}.
\end{aligned}
$$

Plugging $R = \frac{4}{5}\log\left(\frac{n\varepsilon}{\sqrt{d\widetilde{\kappa}\log(1/\delta)}\,\kappa^{5/3}\log^2(n)}\right)$, we finally get

$$\mathbb{E}[\mathsf{Gap}_{F_{\mathcal{D}}}(\widehat{w})] \leq 9D(L_0 + L_1 D)\kappa^{2/3}\frac{d^{1/5}\,\widetilde{\kappa}^{1/5}\log^{1/5}(1/\delta)\log^{4/5}(n)}{n^{2/5}\varepsilon^{2/5}}$$

$$+ 24DL_0\,\kappa^{2/3}\left(\sqrt{\frac{\kappa\log^2(n)}{n}} + \frac{\sqrt{d\kappa\widetilde{\kappa}\log(1/\delta)}\log^2(n)}{n\varepsilon}\right)\frac{d^{1/5}\,\widetilde{\kappa}^{1/5}\log^{1/5}(1/\delta)\log^{4/5}(n)}{n^{2/5}\varepsilon^{2/5}}$$

$$+ 48D(L_0 + L_1 D)\kappa^{2/3}\frac{\varepsilon^{1/5}\log^{3/5}(n)}{n^{3/10}\left(d\widetilde{\kappa}\log(1/\delta)\right)^{1/10}} + 24D(L_0 + L_1 D)\frac{d^{1/5}\,\widetilde{\kappa}^{1/5}\log^{1/5}(1/\delta)\log^{4/5}(n)}{\kappa^{1/2}\,n^{2/5}\,\varepsilon^{2/5}}$$

$$= O\left(D(L_0 + L_1 D)\kappa^{2/3}\left(\frac{\varepsilon^{1/5}\log^{3/5}(n)}{n^{3/10}\left(d\widetilde{\kappa}\log(1/\delta)\right)^{1/10}} + \frac{d^{1/5}\,\widetilde{\kappa}^{1/5}\log^{1/5}(1/\delta)\log^{4/5}(n)}{n^{2/5}\varepsilon^{2/5}}\right)\right).$$

Now, observe that the bound above is dominated by the first term when $d\widetilde{\kappa} = o\left(\frac{n^{1/3}\varepsilon^2}{\log(1/\delta)\log^{2/3}(n)}\right)$. Moreover, note that the first term is decreasing in $d$. Thus, we can obtain a more refined bound via the following simple argument. When $d\widetilde{\kappa} = o\left(\frac{n^{1/3}\varepsilon^2}{\log(1/\delta)\log^{2/3}(n)}\right)$, we embed our optimization problem in higher dimensions; namely, in $d'$ dimensions, where $d'$ satisfies: $d'\left(1 + \log(d') \cdot \mathbf{1}(p < 2)\right) = \Theta\left(\frac{n^{1/3}\varepsilon^2}{\log(1/\delta)\log^{2/3}(n)}\right)$. In such case, the bound above (with $d = d'$) becomes $O\left(D(L_0 + L_1 D)\kappa^{2/3}\frac{\log^{2/3}(n)}{n^{1/3}}\right)$. When $d\widetilde{\kappa} = \Omega\left(\frac{n^{1/3}\varepsilon^2}{\log(1/\delta)\log^{2/3}(n)}\right)$, the bound above is dominated by the second term. Putting these together, we finally arrive at the claimed bound:

$$O\left(D(L_0 + L_1 D)\kappa^{2/3}\left(\frac{\log^{2/3}(n)}{n^{1/3}} + \frac{d^{1/5}\,\widetilde{\kappa}^{1/5}\log^{1/5}(1/\delta)\log^{4/5}(n)}{n^{2/5}\varepsilon^{2/5}}\right)\right).$$

## C  Missing Details of Section 5

For this section, we will occasionally require the use of indicator functions. Given a closed convex set $\mathcal{W}$, we define the (convex) indicator function as

$$\chi_{\mathcal{W}}(w) = \begin{cases} 0 & w \in \mathcal{W} \\ +\infty & w \notin \mathcal{W}. \end{cases}$$

Also recall the definition of the normal cone of $\mathcal{W}$ at point $\overline{w} \in \mathcal{W}$, $\mathcal{N}_{\mathcal{W}}(\overline{w}) = \{p \in \mathcal{W} : \langle p, w - \overline{w}\rangle \leq 0\ \forall w \in \mathcal{W}\}$. The normal cone is the subdifferential of the indicator function: $\mathcal{N}_{\mathcal{W}}(w) = \partial\chi_{\mathcal{W}}(w)$.

### C.1  Background Information on Weakly Convex Functions and their Subdifferentials

**Definition 21.** We say that a function $f : \mathcal{W} \mapsto \mathbb{R}$ is $\rho$-weakly convex w.r.t. norm $\|\cdot\|$ if for all $0 \leq \lambda \leq 1$ and $w, v \in \mathcal{W}$, we have

$$f(\lambda w + (1 - \lambda)v) \leq \lambda f(w) + (1 - \lambda)f(v) + \frac{\rho\lambda(1 - \lambda)}{2}\|w - v\|^2.$$

For nonconvex functions, defining the subdifferential can be done in a local fashion.

**Definition 22.** Let $f : \mathbf{E} \mapsto \mathbb{R}$. We define the *(regular) subdifferential* of $f$ at point $w \in \mathbf{E}$, denoted $\partial f(w)$, as the set of vectors $g \in \mathbf{E}$ such that

$$\liminf_{v \to w, v \neq w}\frac{f(v) - f(w) - \langle g, v - w\rangle}{\|v - w\|} \geq 0.$$

We say that $f$ is subdifferentiable at $w$ if $\partial f(w) \neq \emptyset$. We will say $f$ is subdifferentiable if it is subdifferentiable at every point.

We will need a characterization of the regular subdifferential in terms of directional derivatives. We recall the definition of the directional derivative of a function $f$ at point $w$ in direction $e$:

$$f'(x; e) := \liminf_{\varepsilon \to 0, c \to e} \frac{f(w + \varepsilon e) - f(w)}{\varepsilon}.$$

**Proposition 23** (Regular subdifferential and directional derivatives). *Let $f : \mathbf{E} \mapsto \mathbb{R}$ be a Lipschitz function which is subdifferentiable at $w$, then*

$$\partial f(w) = \{g \in \mathbf{E} : \langle g, e \rangle \leq f'(w; e) \; \forall e \in \mathbf{E}\}.$$

*Proof.* Let $L_0$ be the Lipschitz constant of $f$ w.r.t. $\|\cdot\|$. We prove both inclusions. First ($\subseteq$), if $g \in \partial f(w)$, then let $e \in \mathbf{E} \setminus \{0\}$. Using the definition of subdifferential for $w$ and $v = w + \varepsilon c$ (where $\varepsilon \to 0$ and $c \to e$), we get

$$\liminf_{\varepsilon \to 0, c \to e} \frac{f(w + \varepsilon c) - f(w)}{\varepsilon \|c\|} - \frac{\langle g, c \rangle}{\|c\|} \geq 0$$

Taking first the limit $c \to e$ and then $\varepsilon \to 0$, we get $f'(w; e) \geq \langle g, e \rangle$, concluding the desired inclusion.

For the reverse inclusion ($\supseteq$), let $g \in \mathbf{E}$ be s.t. $\langle g, e \rangle \leq f'(w; e)$, for all $e \in \mathbf{E}$. Now let $v \to w$, and consider any $e \in \mathbf{E}$ accumulation point of $(v - w)/\|v - w\|$ (they exist by compactness of the unit sphere). Next, let $\varepsilon = \|v - w\|$, and notice that $\varepsilon \to 0$. Then

$$
\begin{aligned}
f(v) &= f(w) + [f(v) - f(w + \varepsilon e)] + [f(w + \varepsilon e) - f(w)] \\
&\geq f(w) - L_0 \|(v - w) - \varepsilon e\| + \frac{f(w + \varepsilon e) - f(w)}{\varepsilon} \varepsilon \\
&\geq f(w) + \frac{f(w + \varepsilon e) - f(w)}{\varepsilon} \varepsilon - L_0 \|v - w\| \left( \frac{v - w}{\|v - w\|} - e \right).
\end{aligned}
$$

Taking $v \to w$ (which is equivalent to $\varepsilon \to 0$), we get

$$
\begin{aligned}
f(v) &\geq f(w) + f'(w; e)\varepsilon + o(\|v - w\|) \\
&\geq f(w) + \langle g, \varepsilon e \rangle + o(\|v - w\|) \\
&= f(w) + \langle g, v - w \rangle + \varepsilon \left\langle g, e - \frac{(v - w)}{\varepsilon} \right\rangle + o(\|v - w\|) \\
&= f(w) + \langle g, v - w \rangle + o(\|v - w\|),
\end{aligned}
$$

where in the second step we used the starting assumption. $\square$

Finally, we present the well-known fact that weak convexity implies that the variation of the function compared to its subgradient approximation is lower bounded by a negative quadratic.

**Proposition 24** (Characterization of weak convexity from the regular subdifferential). *Let $f : \mathcal{W} \mapsto \mathbb{R}$ be subdifferentiable and Lipschitz w.r.t. $\|\cdot\|$. Then $f$ is $\rho$-weakly convex if and only if for all $w, v \in \mathbf{E}$, and $g \in \partial f(w)$*

$$f(v) \geq f(w) + \langle g, v - w \rangle - \frac{\rho}{2} \|v - w\|^2. \tag{15}$$

*Proof.* We prove both implications. For $\Rightarrow$, let $v, w \in \mathbf{E}$, and $0 < \lambda < 1$. By $\rho$-weak convexity:

$$f((1 - \lambda)v + \lambda w) \leq (1 - \lambda)f(v) + \lambda f(w) + \frac{\rho\lambda(1 - \lambda)}{2}\|v - w\|^2$$

$$\implies \quad (1 - \lambda)[f(v) - f(w)] \geq f((1 - \lambda)v + \lambda w) - f(w) - \frac{\rho\lambda(1 - \lambda)}{2}\|v - w\|^2$$

$$\implies \quad f(v) - f(w) \geq \lim_{\lambda \to 1} \inf \left[ \frac{f(w + (1 - \lambda)(v - w)) - f(w)}{(1 - \lambda)} - \frac{\rho\lambda}{2}\|v - w\|^2 \right]$$

$$= f'(w; v - w) - \frac{\rho}{2}\|v - w\|^2$$

$$\geq \langle g, v - w \rangle - \frac{\rho}{2}\|v - w\|^2,$$

where in the last inequality we used Proposition 23.

Next, for $\Leftarrow$, let $v, w \in \mathbf{E}$ and $0 \leq \lambda \leq 1$. Then, letting $g \in \partial f((1 - \lambda)w + \lambda v)$, and using (15) twice, we get

$$f(v) \quad \geq \quad f((1 - \lambda)w + \lambda v) + \langle g, (1 - \lambda)(v - w) \rangle - \frac{\rho}{2}\|(1 - \lambda)(v - w)\|^2$$

$$f(w) \quad \geq \quad f((1 - \lambda)w + \lambda v) + \langle g, \lambda(w - v) \rangle - \frac{\rho}{2}\|\lambda(v - w)\|^2.$$

Multiplying the first inequality by $\lambda$ and the second one by $(1 - \lambda)$, gives

$$\lambda f(v) + (1 - \lambda)f(w) \quad \geq \quad f((1 - \lambda)w + \lambda v) - \frac{\rho\lambda(1 - \lambda)}{2}\|v - w\|^2,$$

which concludes the proof. $\qquad \square$

From the previous proposition, we can easily conclude that any smooth function is weakly convex.

**Corollary 25.** *Let $f : \mathcal{W} \mapsto \mathbb{R}$ be a $L_1$-smooth function (i.e., $\|\nabla f(v) - \nabla f(w)\|_* \leq L_1\|v - w\|$, for all $v, w \in \mathcal{W}$). Then $f$ is $L_1$-weakly convex.*

*Proof.* Let $v, w \in \mathcal{W}$. Then by the Fundamental Theorem of Calculus:

$$f(v) = f(w) + \int_0^1 \langle \nabla f(w + s(v - w)), v - w \rangle ds$$

$$= f(w) + \langle \nabla f(w), v - w \rangle + \int_0^1 \langle \nabla f(w + s(v - w)) - \nabla f(w), v - w \rangle ds$$

$$\geq f(w) + \langle \nabla f(w), v - w \rangle - L_1\|v - w\|^2 \int_0^1 s ds.$$

We conclude by Proposition 24 that $f$ is $L_1$-weakly convex. $\qquad \square$

### C.1.1 Basic Rules of the Subdifferential, Optimality Conditions and Stationarity Gap

We know provide some basic tools regarding subdifferentials and optimality conditions in weakly convex programming, which will also allow us to introduce the notion of stationarity gap in this setting.

To start, we provide a basic calculus rule for the subdifferential of a sum of weakly convex functions.

**Theorem 26** (Corollary 10.9 from [RW98]). *If $f : \mathbf{E} \mapsto \mathbb{R}$ be weakly convex, and $g : \mathbf{E} \mapsto \mathbb{R} \cup \{+\infty\}$ be convex, lower semicontinuous, and such that $w \in dom(g)$. Then $\partial(f + g)(w) = \partial f(w) + \partial g(w)$.*

Next, we provide a relation between directional derivatives and the regular subdifferential.

**Proposition 27** (From Proposition 8.32 in [RW98]). *If $\varphi : \mathbb{E} \mapsto \mathbb{R} \cup \{+\infty\}$ is weakly convex, then*

$$\text{dist}(0, \partial\varphi(w)) = -\inf_{\|e\| \leq 1} \varphi'(w; e).$$

With these results, we can now provide optimality conditions for weakly convex optimization

**Proposition 28** (Stationarity conditions for weakly convex optimization). *Let $f : \mathcal{W} \mapsto \mathbb{R}$ be $\rho$-weakly convex and $L_0$-Lipschitz w.r.t. $\|\cdot\|$, and $\mathcal{W}$ a closed and convex set. Then, if $w^* \in \arg\min\{f(w) : w \in \mathcal{W}\}$, then there exists $g \in \partial f(w^*)$ such that*

$$\langle g, v - w^* \rangle \geq 0 \qquad (\forall v \in \mathcal{W}).$$

*Proof.* First, we observe that without loss of generality, $f : \mathbb{E} \mapsto \mathbb{R}$ (this is a consequence of the Lipschitz extension Theorem). Let now $g(w) = \chi_{\mathcal{W}}(w)$ (i.e., the convex indicator function, as defined in the beginning of this section). Since $w^* \in \mathcal{W}$, by Proposition 26, we have $\partial(f + g)(w^*) = \partial f(w^*) + \partial g(w^*)$. Now we apply Proposition 28 to $\varphi(w) = f(w) + g(w)$; since $w^*$ is a minimizer of $\varphi$, we have that $\varphi'(w^*; e) \geq 0$ for all $e$, and hence $\text{dist}(0, \partial\varphi(w^*)) = 0$. Since $\partial g(w^*) = \mathcal{N}(w^*)$, we get that

$$0 = \text{dist}(0, \partial f(w^*) + \mathcal{N}_{\mathcal{W}}(w^*)),$$

and this implies that there exists $g \in \partial f(w^*)$, such that $g \in -\mathcal{N}_{\mathcal{W}}(w^*)$, i.e.,

$$\langle g, v - w^* \rangle \geq 0 \qquad (\forall v \in \mathcal{W}).$$

$\square$

The previous result leads to a natural definition of the stationarity gap in weakly convex optimization:

$$\mathsf{Gap}_f(w) = \inf_{g \in \partial f(w)} \sup_{v \in \mathcal{W}} \langle g, v - w \rangle. \tag{16}$$

Notice that, by Proposition 28, any minimizer of a weakly convex and Lipschitz function is such that its stationarity gap is equal to zero.

## C.2 Missing proofs from Section 5.1

### C.2.1 Proof of Proposition 13

By strong convexity of $\frac{1}{2}\|\cdot\|^2$ and weak convexity of $f$:

$$\frac{\beta}{2}\|[\lambda w + (1-\lambda)v] - u\|^2 \leq \lambda\frac{\beta}{2}\|w - u\|^2 + (1-\lambda)\frac{\beta}{2}\|v - u\|^2 - \frac{\beta\nu\lambda(1-\lambda)}{2}\|w - v\|^2$$

$$f(\lambda w + (1-\lambda)v) \leq \lambda f(w) + (1-\lambda)f(v) + \frac{\rho\lambda(1-\lambda)}{2}\|w - v\|^2$$

Adding these inequalities, and using that $\nu\beta \geq \rho$, we conclude the $(\nu\beta - \rho)$-strong convexity of $f(\cdot) + \frac{\beta}{2}\|\cdot -u\|^2$, concluding the proof.

### C.2.2 Proof of Lemma 14

We now present the proof. First, notice that the proximal-type mapping can be computed as a solution of the optimization problem

$$\min_{v \in \mathcal{W}} \left[f(v) + \frac{\beta}{2}\|v - w\|^2\right] \tag{17}$$

By Proposition 13, problem (17) is strongly convex, and therefore it has a unique solution; in particular, $\hat{w}$ is well-defined and unique. Next, we use the optimality conditions of constrained convex optimization for problem (17), together with the subdifferential of the sum rule (Theorem 26), and the chain rule of the convex subdifferential; to conclude that

$$\Big(\partial f(\hat{w}) + \beta\|\hat{w} - w\|\,\partial(\|\cdot\|)(\hat{w} - w)\Big) \cap -\mathcal{N}_{\mathcal{W}}(\hat{w}) \neq \emptyset. \tag{18}$$

First, consider the case where $\hat{w} = w$, then there exists $g \in \partial f(\hat{w})$ s.t., $\langle g,\, \hat{w} - v\rangle \leq 0$, for all $v \in \mathcal{W}$, which shows the desired conclusion. In the case $\hat{w} \neq w$, consider $g \in \partial f(\hat{w})$ and $h \in \partial(\|\cdot\|)(\hat{w} - w)$ such that by (18), $\langle g + \beta\|\hat{w} - w\|h, v - \hat{w}\rangle \geq 0$, for all $v \in \mathcal{W}$. We first prove that $\|h\|_* = 1$. Indeed, first $\|h\|_* \leq 1$ since the norm is 1-Lipschitz. The reverse inequality follows from the equality in the Fenchel inequality, when $p$ is a subgradient [HUL01],

$$\|\hat{w} - w\| = \|\hat{w} - w\| + \chi_{\mathcal{B}_*(0,1)}(p) = \langle p, \hat{w} - w\rangle.$$

Since $\hat{w} \neq w$, this shows in particular that $\|p\|_* = 1$. We conclude that in this case, $\langle g, \hat{w} - v\rangle \leq \beta D\|w - \hat{w}\|$, for all $v \in \mathcal{W}$, which concludes the proof.

### C.2.3 Missing Details in Consequences of Proximal Near Stationarity

Now we explain some technical details behind the derivation of the following consequence for proximal nearly-stationary algorithms

$$\mathbb{E}_{S\sim\mathcal{D}^n,\mathcal{A}}\Big[\|\mathsf{prox}_{F_\mathcal{D}}^\beta(\mathcal{A}(S)) - \mathcal{A}(S)\|\Big] \leq \vartheta \qquad \text{and} \qquad \mathbb{E}_{S\sim\mathcal{D}^n,\mathcal{A}}\Big[\mathsf{Gap}_{F_\mathcal{D}}\big(\mathsf{prox}_{F_\mathcal{D}}^\beta(\mathcal{A}(S))\big)\Big] \leq \vartheta. \tag{19}$$

First, we suppose $\mathcal{A}$ is $(\vartheta, \beta)$-proximal nearly stationary. From this, we directly conclude the first property,

$$\mathbb{E}_{S\sim\mathcal{D}^n,\mathcal{A}}\Big[\|\mathsf{prox}_{F_\mathcal{D}}^\beta(\mathcal{A}(S)) - \mathcal{A}(S)\|\Big] \leq \vartheta.$$

For the second property, we first recall the stationarity gap in weakly convex optimization (see eqn. (16)): here, for $w \in \mathcal{W}$ and objective $f : \mathcal{W} \mapsto \mathbb{R}$, define

$$\mathsf{Gap}_f(w) = \inf_{g\in\partial f(w)} \sup_{v\in\mathcal{W}} \langle g, w - v\rangle.$$

Now, if $\mathcal{B} : \mathcal{Z}^n \mapsto \mathbb{R}$ is a randomized algorithm, its expected gap corresponds to

$$\mathbb{E}_{S\sim\mathcal{D}^n,\mathcal{B}}[\mathsf{Gap}_{F_\mathcal{D}}(\mathcal{B}(S))] = \mathbb{E}_{S\sim\mathcal{D}^n,\mathcal{B}}\Big[\inf_{g\in\partial F_\mathcal{D}(\mathcal{B}(S))} \sup_{v\in\mathcal{W}} \langle g, \mathcal{B}(S) - v\rangle\Big].$$

Finally, under this definition of the expected gap, we have that if $\mathcal{B}(S) = \mathsf{prox}_{F_\mathcal{D}}^\beta(\mathcal{A}(S))$, then by Lemma 14 and $(\vartheta, \beta)$-proximal near stationarity,

$$\mathsf{Gap}_{F_\mathcal{D}}(\mathcal{B}) = \mathbb{E}_{S\sim\mathcal{D}^n,\mathcal{B}}\Big[\inf_{g\in\partial F_\mathcal{D}(\mathcal{B}(S))} \sup_{v\in\mathcal{W}} \langle g, \mathcal{B}(S) - v\rangle\Big] \leq \mathbb{E}_{S\sim\mathcal{D}^n,\mathcal{B}}\Big[\beta D\|\mathcal{B}(S) - \mathcal{A}(S)\|\Big]$$

$$\leq \vartheta,$$

concluding the claim.

### C.3 Missing Details of Section 5.2

Algorithm 2 is inspired by the *proximally guided stochastic subgradient method* of Davis and Grimmer [DG19], where the proximal subproblems are solved using an optimal algorithm for DP-SCO in the strongly convex case, given in [AFKT21]. Hence, we cite their theorem below.

**Theorem 29** (Thm. 8 in [AFKT21]). *Consider the $\ell_p$ setting of $\lambda$-strongly convex stochastic optimization, where $1 \leq p \leq 2$. There exists an $(\varepsilon, \delta)$-differentially private algorithm $\mathcal{A}_{\mathrm{SC}}$ with excess risk*

$$O\Big(\frac{L_0^2}{\lambda}\Big[\frac{\kappa}{n} + \frac{\tilde{\kappa}\kappa^2 d\log(1/\delta)}{n^2\varepsilon^2}\Big]\Big),$$

*where $\kappa = \min\{1/(p-1), \log d\}$ and $\tilde{\kappa} = 1 + \log d \cdot \mathbf{1}(p < 2)$. This algorithm runs in time $O(\log n \cdot \log\log n \cdot \min\{n^{3/2}\sqrt{\log d}, n^2\varepsilon/\sqrt{d}\})$.*

We note in passing that Thm. 8 in [AFKT21] is stated only for the $\ell_1$-setting; however, since their mirror descent algorithm and reduction to the strongly convex case works more generally, we state a more general version of their statement.

### C.3.1 Proof of Theorem 16

The privacy of Algorithm 2 is certified by parallel composition and the privacy guarantees of $\mathcal{A}_{SC}$. For the accuracy, first consider the case $p \geq 1 + 1/\log d$. Here, recall that $w \mapsto \frac{1}{2}\|w - \bar{w}\|_p^2$ is a $1/\kappa$-strongly convex function w.r.t. $\|\cdot\|_p$ [Bec17], so we can choose $\nu = 1/\kappa = (p-1)$ as the strong convexity parameter. Let $\hat{w}_r = \text{prox}_{F_{\mathcal{D}}}^{\beta}(w_r)$ be the optimal solution to problem (6). Our goal now is to show that $\overline{w}^R$ is $\vartheta$-proximal nearly stationary. First, by Proposition 13, $F_r$ is $(\beta/\kappa - \rho)$-strongly convex w.r.t. $\|\cdot\|_p$. Since $(\beta/\kappa - \rho) = \rho$, we have by Theorem 29 that for all $r = 1, \ldots, R$,

$$\mathbb{E}\big[F_r(w_{r+1}) - F_r(\hat{w}_r)\big] = O\Big(\frac{L_0^2}{\rho}\Big[\frac{\kappa}{n_r} + \frac{\tilde{\kappa}\kappa^2 d \log(1/\delta)}{n_r^2 \varepsilon^2}\Big]\Big). \tag{20}$$

By strong convexity of $F_r$, we have almost surely:

$$F_{\mathcal{D}}(w_r) = F_r(w_r) \geq F_r(\hat{w}_r) + \frac{\rho}{2}\|\hat{w}_r - w_r\|_p^2. \tag{21}$$

Hence, using (20) and (21), we get

$$\mathbb{E}\Big[F_{\mathcal{D}}(w_{r+1}) + \frac{\beta}{2}\|w_{r+1} - w_r\|_p^2\Big] = \mathbb{E}[F_r(w_{r+1})] \leq \mathbb{E}[F_r(\hat{w}_r)] + O\Big(\frac{L_0^2}{\rho}\Big[\frac{\kappa}{n_r} + \frac{\tilde{\kappa}\kappa^2 d \log(1/\delta)}{n_r^2 \varepsilon^2}\Big]\Big)$$

$$= \mathbb{E}\Big[F_{\mathcal{D}}(w_r) - \frac{\rho}{2}\|\hat{w}_r - w_r\|_p^2\Big] + O\Big(\frac{L_0^2}{\rho}\Big[\frac{\kappa}{n_r} + \frac{\tilde{\kappa}\kappa^2 d \log(1/\delta)}{n_r^2 \varepsilon^2}\Big]\Big),$$

and summing from $r = 1, \ldots, R$, we obtain

$$\frac{1}{R}\sum_{r=1}^{R} \mathbb{E}\|\hat{w}_r - w_r\|^2 \leq \frac{2}{R\rho}\Big[\mathbb{E}[F(w_1) - F(w_{R+1})] + O\Big(\sum_{r=1}^{R} \frac{L_0^2}{\rho}\Big[\frac{\kappa}{n_r} + \frac{\tilde{\kappa}\kappa^2 d \log(1/\delta)}{n_r^2 \varepsilon^2}\Big]\Big)\Big]$$

$$= O\Big(\frac{1}{\rho}\Big\{\frac{L_0 D}{R} + \frac{L_0^2}{\rho}\Big[\kappa\frac{R}{n} + \frac{\tilde{\kappa}\kappa^2 d \log(1/\delta)}{\varepsilon^2}\frac{R^2}{n^2}\Big]\Big\}\Big).$$

Now we use that $R = \Big\lfloor \min\Big\{\sqrt{\frac{nD\rho}{\kappa L_0}}, \frac{1}{(\tilde{\kappa}\kappa^2)^{1/3}}\big(\frac{D(n\varepsilon)^2\rho}{L_0 d \log(1/\delta)}\big)^{1/3}\Big\}\Big\rfloor$, which is at most $n$ by the assumption $nd \geq \rho D/L_0$. Then,

$$\mathbb{E}\big[\|\text{prox}_{F_{\mathcal{D}}}(\overline{w}^R) - \overline{w}^R\|_p^2\big] = \frac{1}{R}\sum_{r=1}^{R} \mathbb{E}\big[\|\hat{w}_r - w_r\|_p^2\big] = O\left(\frac{1}{\rho}\Big[\frac{L_0^{3/2}D\sqrt{\kappa}}{\sqrt{n\rho}} + (\tilde{\kappa}\kappa^2)^{1/3}(L_0^2 D)^{2/3}\Big(\frac{d \log(1/\delta)}{(n\varepsilon)^2\rho}\Big)^{1/3}\Big]\right).$$

Finally, by the Jensen inequality, we have that

$$\mathbb{E}\big[\max\{1, \beta D\}\|\text{prox}_{F_{\mathcal{D}}}(\overline{w}^R) - \overline{w}^R\|_p\big] \leq \frac{\max\{1, 2\rho D\kappa\}}{\sqrt{\rho}}O\Big(\frac{L_0^{3/2}(D\kappa)^{1/4}}{[n\rho]^{1/4}} + (\tilde{\kappa}\kappa)^{1/6}(L_0^2 D)^{1/3}\Big(\frac{d \log(1/\delta)}{(n\varepsilon)^2\rho}\Big)^{1/6}\Big).$$

Next, in the case $1 \leq p < 1 + 1/\log d$, we can use that $\|\cdot\|_{\bar{p}}$ and $\|\cdot\|_p$ are equivalent with a constant factor (recall that here $\bar{p} = 1 + 1/\log d$). Using then $\|\cdot\|_{\bar{p}}$ in the algorithm and argument above clearly leads to the same conclusion with $\kappa = \log d$. Finally, the running time upper bound follows by Theorem 29.