# OpenReview forum: "Differentially Private Stochastic Optimization: New Results in Convex and Non-Convex Settings"
_NeurIPS.cc/2021/Conference — NeurIPS 2021 Poster_

### Official Review · Reviewer_Brcd · 2021-07-15

**Rating:** 7
**Confidence:** 2

**Summary:**

This work studies differentially private stochastic optimization and proposes several new algorithms for convex non-smooth generalized linear losses (GLL) / general non-convex smooth objectives / weakly convex non-smooth objectives equipped with Euclidean / non-Euclidean geometry.

**Limitations And Societal Impact:**

The authors have adequately addressed the limitations and potential negative societal impact of their work.

**Main Review:**

This work proposes several new algorithms that outperform the existing methods in differentially private stochastic optimization. The authors leverage techniques such as Moreau envelop smoothing for GLL and variance-reduced stochastic Frank-Wolfe to build several linear-time algorithms in the DP setting. The authors also make the first attempt to devise algorithms for weakly convex non-smooth objectives in the DP setting. The contributions are significant and the authors are rather careful in presenting the theoretical results, which is highly appreciated. The paper is well-written but with a lot of proposed algorithms and technical details deferred to the appendix, the results remain hard to parse. Perhaps it suggests that a conference with a small page limit is not the best venue?

**Time Spent Reviewing:**

4

---

> ### Author Response · Authors · 2021-08-08
> **Initial Response**
>
> We thank the reviewer for the positive assessment of the paper. Regarding the comment *“The paper is well-written but with a lot of proposed algorithms and technical details deferred to the appendix, the results remain hard to parse. Perhaps it suggests that a conference with a small page limit is not the best venue?”* we would like to say that many important theoretical papers in NeurIPS are structured in this way: the main body of the paper contains an extended outline of the results with discussions on the most novel aspects of the work, and detailed proofs are deferred to the Appendix. Therefore, we believe the paper should not be disregarded for proofs’ length considerations.

---

### Official Review · Reviewer_c6Mx · 2021-07-17

**Rating:** 4
**Confidence:** 4

**Summary:**

The paper studies three different (slightly related) problems in private optimization: 1. convex private optimization for non-smooth functions, 2. non-convex smooth private optimization, and 3. non-convex non-smooth private optimization. The authors study each of these settings separately, developing different algorithms for each that has certain advantages. The nature of improvement is different in each setting: in the first, the author claim to improve runtime (achieving linear runtime) while maintaining statistical utility, in the second, (for ell_2 geometry) the authors achieve worse rates while getting better runtime, and in the third setting the authors consider the setting of weakly-convex non-smooth functions (which I think was not studied before in the privacy literature) and provide private algorithms with certain convergence guarantees.

**Limitations And Societal Impact:**

Yes

**Main Review:**

I think the results are mostly incremental, based on standard techniques with limited novelty (see below). Moreover, the different problems (settings) in the paper are only slightly related, making the story of the paper less coherent: basically a list of different results that the author prove in different settings. Overall, I don't think the paper has sufficient novelty and new results for acceptance.


1. Contribution for non-smooth GLL: while it is true that existing algorithms from [BFTT19, FKT20] take super-linear time for general non-smooth convex functions, the main bottleneck there is smoothening general functions which takes linear time. However, for the setting of GLL that the authors consider, it is immediate to see that existing algorithms give nearly linear time algorithms as smoothening GLLs (or finding the gradient of Moreau envelope) is quite simple as it is basically a 1-dimensional problem. So this result is basically a direct corollary of previous work.

2. Comments on smooth non-convex setting:

    2.1 The results for this section are based on private implementation of variance-reduced Frank-Wolfe algorithm similarly to the
    existing private algorithm for the convex setting [AFKT21, BGN21]. The algorithm in this paper doesn't use privacy amplification by
    sub-sampling (as done in [AFKT21]) hence suggesting their algorithm may not be optimal.

    2.2 The author conjecture (line 78) that their algorithm has tight bounds for the ell_1 setting for linear time algorithms. Is there
    any evidence for this prediction? Following my previous comment (2.1), I think it may be possible to improve the rates using better
    privacy accounting as done in [AFKT21].

    2.3 Overall the authors overuse the term: "we provide the first algorithm for ...".  One such example is line 76 where the authors
    say "we provide the first linear time private algorithms". There are two problems here. First, this sentence doesn't say anything
    about the utility of the algorithm and so is not very informative. Secondly, the linear-time algorithm doesn't give optimal bounds
    as [ZCH+20]. While the algorithm of [ZCH+20] takes n^2/\sqrt{d} time, we can run it for linear number of steps if we require a
    linear time algorithm. The authors should compare their bound to the bound obtained by linear-time version of [ZCH+20].


3. Non-smooth weakly-convex: the authors should try to explain the strength of the bounds achieved in this section. Currently these are worse than the smooth setting which suggest they may not be optimal as the bounds are usually similar in ell_2 case for both smooth and non-smooth convex case. Moreover, the algorithm is a simple application of existing techniques.


More comments:

1. I think it would be useful to add the dependence on epsilon in the bounds of the abstract as currently these are stated only for constant epsilon and makes it slightly harder to understand the privacy cost.

2. The term GLM (generalized linear models) is more common than GLL.



**Time Spent Reviewing:**

2-4 hours

---

> ### Author Response · Authors · 2021-08-08
> **Initial Response**
>
> We disagree with the assessment of this work being incremental. First, this comment ignores several novelties in this work: existing work on DP nonconvex stochastic optimization has been exclusively done in unconstrained, smooth and Euclidean cases. Our results are the first to remove these assumptions. Second, although our methods build upon past work, to obtain non-trivial convergence guarantees in the nonconvex setting under DP, the batch allocation, stepsize policy and sensitivity analysis have to be completely re-designed (in fact, using the choices made in the convex private case or in the non-convex non-private case results in much weaker or uninformative convergence guarantees). Finally, given the interest by the NeruIPS community in stochastic nonconvex optimization, and the limitations of the current approaches for this problem under DP, we believe this work fills an important gap.
> Please see detailed responses below:
> 1. Regarding the comment on smoothing of GLLs, we agree this is based on a simple idea. This is not the core of our results, but we believe it is worth stating given recent refinements in the running time of DP-SCO and that recent work such [SSTT21] have demonstrated the benefits of GLLs in the private setting w.r.t. excess risk, but not running time. Further, as pointed out by reviewer MuL7, in the private setting the fact that linear time optimization is possible for GLLs indicates that fundamentally different constructions are needed for lower bounding oracle complexity than those used to lower bound excess risk. We would also contend that the realization that such smoothing allows for better rates in the $\ell_1$ setting is far less obvious, as general smoothings in this case do not exist.
>
> 2. Response for smooth nonconvex setting:
>     1. In AFKT21, privacy amplification by shuffling is shown to be useful only in the high privacy regime, namely $\varepsilon=\tilde{O}(1/n^{1/4})$ (see Thm. 7 in https://arxiv.org/pdf/2103.01516.pdf). For $\varepsilon=\Theta(1)$, it does not lead to improvements. We nevertheless investigated the technique of [AFKT21] for the nonconvex case and didn’t obtain any improvements, even in the high privacy regime: one reason for this is that batch schedules in the nonconvex case are quite different from the convex one (in AFKT’s tree-based algorithm batch sizes decrease exponentially fast which leads to highly suboptimal accuracy in the nonconvex case). It is not safe to assume that what works in the convex case will work in the non-convex case.
>     2. Our claim of optimality is only indirect, and comes from an oracle lower bound for unconstrained $\ell_2$ setting: there a lower bound of $\Omega(1/n^{1/3})$ was proved (Arjevani et al. 2019,https://arxiv.org/abs/1912.02365). There are several caveats, including whether this lower bound could also apply to $\ell_1$, and whether constraining the feasible set can still make their argument work. But we are not aware of any case in (nonprivate) stochastic optimization where the $\ell_1$ case is easier than the $\ell_2$ case, or where constrained is easier than unconstrained.
>     3. To the best of our knowledge, our claims are factual. If you believe we are wrong, please let us know what reference we are missing. Also note that although [ZCH+20] obtains slightly better sample complexity, their algorithm has worse oracle complexity. Specifically, to obtain an $\alpha$ approximate stationary point, [ZCH+20] achieves oracle complexity $\alpha^{-4}\sqrt{d}$ whereas our algorithm achieves $\max(\alpha^{-5/2}\sqrt{d}, \alpha^{-10/3}d^{-1/3}$)  (moreover, we have recently been able to further refine this bound to $\max\{\alpha^{-5/2}\sqrt{d},\alpha^{-3}\}$). Additionally, [ZCH+20] considers the unconstrained setting, whereas we consider the constrained setting. With regards to your other point, due to the fact that the algorithm in [ZCH+20] relies on full batch gradients, a linear time version would only run for a constant number of iterations, and thus yield trivial accuracy. Finally, [ZCH+20] is not known to be optimal, as there are no private lower bounds in this setting.
> 3. The existing bounds for the non-smooth weakly convex setting do in fact differ from existing bounds for the smooth setting. For non-convex nonsmooth stochastic optimization, the state of the art is $O(1/n^{1/4})$, achieved by Davis et al. in a series of papers ([DD19,DG19] https://arxiv.org/abs/1707.03505, https://arxiv.org/abs/1803.06523, etc.). There is also a matching lower bound for oracle based algorithms by Arjevani et al. [ACD+19], so it is likely our bounds are unimprovable (at least for some ranges of parameters). Here again, what is known for the convex case does not translate directly to the non-convex one.
>
> For the last two points
> 1. We will also add the dependence on $\varepsilon$ to the table of results.
> 2. The term GLL has been used by papers such as https://arxiv.org/abs/1812.06825. It is more appropriate to our analysis, which hinges on the form of the loss function and is distribution independent.

---

### Official Review · Reviewer_MuL7 · 2021-07-18

**Rating:** 8
**Confidence:** 3

**Summary:**

The paper considers differential private stochastic optimization in a number of convex and non-convex settings. Their contributions are as follows:

1. Faster (nearly linear time) algorithm for $\ell_2$ non-smooth convex generalized linear losses (GLLs).

2. Algorithm with smaller error (nearly dimension independent) in $\ell_1$ non-smooth convex GLLs.

3. New algorithms for smooth non-convex and non-smooth weakly convex stochastic optimization in non-Euclidean ($\ell_p$) settings.

**Limitations And Societal Impact:**

No direct societal impact.

**Main Review:**

1. The paper makes many novel and important contributions to the field of differentially private stochastic optimization. Importantly, as mentioned in the paper, many results in non-Euclidean non-convex settings are novel even without privacy constraints.

2. The writing and presentation is very good and to the point. Also, the authors try to give the necessary background (in the Appendix) and explain many subtleties as required.

3. The first result is a nearly-linear time algorithm for non-smooth GLLs. This contributes to a recent line of work which try to devise fast algorithms for DP-SCO. This result solves a special case of the non-smooth convex landscape. I think this result has an interesting consequence: *most* known high-dimensional oracle complexity lower bound instances in optimization are GLLs. So this result tells us that if it is indeed the case that non-smooth DP-SCO is harder than SCO, then we need to look at *new* hard instances.

4. The smooth non-convex setting considers a polyhedral constrained setup even in standard $\ell_2$ setting. I think that it is more natural/standard in literature to consider an $\ell_2$ ball (or a simple convex set with an easy oracle-access projection). I wonder why did the authors not consider the later? Also, why not consider the unconstrained setup, which would also enable comparison with some of the previous work?

5. On a related note, the non-smooth weakly convex part is not in the polyhedral constraint setting. So, do we need to access to a projection oracle for the claimed guarantees in Theorem 16? If yes, then I could not find it mentioned in this section. If no, then why is it so?

6. **Coherence**: Since the authors make many contributions, one downside is that parts of the paper feel disconnected from each other -- it feels like a collection of results the authors were able to obtain in various settings. For example: GLLs part of the paper has little to do with the non-convex setting (in terms of goal/techniques). This, lots of settings and results, also led to authors' being able to put only one algorithm description (in full, as a pseudo-code) in the main paper, and the rest is deferred to the appendix.

7. **Tightness of rates**: The authors only provide upper bounds for the settings they consider. Is there a way (perhaps from previous works?) to assess how sharp are the attained rates?


**Time Spent Reviewing:**

4

---

> ### Author Response · Authors · 2021-08-08
> **Initial Response**
>
> We would like to thank the reviewer for the valuable feedback, particularly regarding point 3. One of our motivations to study DP-GLL was precisely to better understand the limits and possibilities of linear time algorithms for DP-SCO with nonsmooth losses. This may not be apparent in the current presentation of the paper, so we will make sure to include this observation in the final version. For other questions and comments, please find the corresponding responses below:
>
> 4.  Our results in Section 4.2 can be used for general domains endowed with a linear optimization oracle, in particular for the setting described by the reviewer. Our assumption of $\ell_p$ ball domain is only made to have a clean bound. For Section 4.1 the polyhedrality assumption is key: this leads to the dimension independent bounds (even tiny deviations from polytopes may lead to dimension-dependent lower bounds, as shown in [BGN21]).
>
> 5. We apologize for the confusion: The projection assumption is implicit. Computing the proximal-type operator (3) can be seen as a type of projection (it indeed requires projections in the $\ell_2$ case). We will add a clarification about this.
>
>   6 & 7. We will move some of the main body into the appendix in order to provide a more intuitive explanation of the main results as well as comparison to other works. Regarding the tightness of our rates, note that lower bounds on the sample complexity of approximating stationary points are only known in the convex case [Foster et al. 2019: https://arxiv.org/pdf/1902.04686.pdf]. For stochastic nonconvex optimization, the only existing lower bounds for approximating stationary points are on the oracle complexity of oracle-based methods [Arjevani et al. 2019: https://arxiv.org/pdf/1912.02365.pdf]: these lower bounds are  usually very loose in the DP case. Moreover, these lower bounds hold only when the dimensions are sufficiently large, and only apply to the Euclidean setting.

---

### Official Review · Reviewer_TpjV · 2021-07-23

**Rating:** 6
**Confidence:** 4

**Summary:**

This paper studied the problem of differentially private stochastic optimization. The authors establish new results for the convex generalized linear model and new algorithms for nonconvex optimization. The main contribution of the current paper is the faster algorithms and stronger results for solving the nonsmooth DP-SCO in the case of generalized linear loss.

**Limitations And Societal Impact:**

Yes

**Main Review:**

I think the new algorithms are interesting and the utility and convergence results seem to be strong. However, my major concerns are about the algorithms and results in nonconvex setting, and I summarize it as follows:
1. It is unclear the novelty of the proposed Private Frank-Wolfe algorithm. It seems to be a straightforward application of the variance reduced stochastic Frank-Wolfe algorithm to the private setting. The authors should clarify the novelty.
2. What are the main challenges for analyzing your proposed Private Frank-Wolfe algorithm? This could help readers understand the contributions of the paper.
3. The proposed Private Frank-Wolfe algorithm requires a very large batch at each iteration, suggesting that the proposed algorithm will have very bad scalability.
4. The utility guarantee of the proposed algorithm is worse than existing work. For example, in the l_2 setting, the utility guarantee is worse than the result in [ZCH+20].
5. There are no experiments to validate the effectiveness of the proposed methods in the current paper.

**Time Spent Reviewing:**

10

---

> ### Author Response · Authors · 2021-08-08
> **Initial Response**
>
> Please see the corresponding responses below:
> 1. Our work is the first to study constrained stochastic non-convex optimization under DP. In such setting, bounding the stationarity gap under DP requires a new analysis that does not follow from any of the existing works; in particular, our analysis entails structuring the algorithms in rounds of increasing length together with balancing the step sizes and minibatches in a way that ensures that the variance of the gradient estimates and sensitivity are carefully controlled across all iterations.
> 2. For our level of generality, the use of regular norms is key: this allows us, for instance, to obtain the first $O(1/n^{1/3})$ rate for $\ell_1$ smooth non-convex setup (privately or non-privately). Related to the point above, bounding sensitivity and balancing it with convergence is one of the main challenges.
> 3. We agree large minibatches might be disadvantageous for large-scale settings. However, our algorithm is single-pass, and thus has linear oracle complexity with respect to the dataset size. Further, this algorithm is projection free, which is highly favorable for practical implementations.
> 4. It is true that the excess risk obtained by our noisy-SFW algorithm is slightly worse, but it is the first linear time algorithm in this setting with a nontrivial accuracy guarantee. Further, [ZCH+20] has a full-batch schedule, which is impractical according to your previous comment. By comparison, our algorithm makes only $n$ gradient evaluations in total, and it is projection-free. Finally, [ZCH+20] does not consider non-euclidian settings.
> 5. This is outside the scope of our paper. Our primary contribution is theoretical and we believe that the fundamental nature of the questions we address here makes the work interesting for the community.

---

### Decision · Program_Chairs · 2021-09-28

**Decision:**

Accept (Poster)

**Comment:**

This paper generated a lot of discussion among reviewers. The authors would be advised to improve the clarity of the discussion in order to better integrate their diverse-seeming results into one story. Please also include the improvements suggested in discussions between the reviewers and the authors.


**Consistency Experiment:**

NeurIPS has a long history of experimentation. In 2014, NeurIPS ran an experiment in which 10% of submissions were reviewed by two independent committees to quantify the randomness in the review process. This year, we repeated a variant of this experiment to see how the quality of the review process has changed over time.  This paper was part of the experiment and was therefore assigned to two committees (consisting of reviewers, an Area Chair, and a Senior Area Chair) that reached independent decisions.  If both committees made the same recommendation, this recommendation was followed. If a single committee recommended acceptance, the paper was accepted (with the exception of a few cases in which the other committee identified what we considered a fatal flaw, e.g., an error in a key result).

This copy’s committee reached the following decision: **Accept (Poster)**

The other committee assigned to the paper recommended **Accept (Spotlight)**.  You can find the other set of reviews, along with any follow up discussion with the authors here:
https://openreview.net/forum?id=Ra-2OvXr7UU